# AltiMaP: Altimetry Mapping Procedure for Hydrography Data

Menaka Revel[1], Xudong Zhou[1], Prakat Modi[1,2], Jean-François Cretaux[3], Stephane Calmant[4], and Dai Yamazaki[1]

[1]Global Hydrological Prediction Center, Institute of Industrial Science, The University of Tokyo, Tokyo, Japan.

[2]Department of Civil Engineering, Shibaura Institute of Technology, Tokyo, Japan.

[3]Laboratoire d'Études en Géophysique et Océanographie Spatiales (LEGOS), Centre National d'Études Spatiales (CNES) Toulouse, France.

[4]Institute of Research for Development, France.

*Correspondence to*: Menaka Revel (menaka@rainbow.iis.u-tokyo.ac.jp)

**Abstract**

Satellite altimetry data are useful for monitoring water surface dynamics, evaluating and calibrating hydrodynamic models, and enhancing river-related variables through optimization or assimilation approaches. However, comparing simulated water surface elevations (WSEs) using satellite altimetry data is challenging due to the difficulty of correctly matching the representative locations of satellite altimetry virtual stations (VSs) to

the discrete river grids used in hydrodynamic models. In this study, we introduce an automated altimetry mapping procedure (AltiMaP) that allocates VS locations listed in the HydroWeb database to the Multi-Error Removed Improved Terrain Hydrography (MERIT Hydro) river network. Each VS was flagged according to the land cover of the initial pixel allocation, with 10, 20, 30, and 40 representing river channel, land with the nearest single-channel river, land with the nearest multi-channel river, and ocean pixels, respectively. Then, each VS was

assigned to the nearest MERIT Hydro river reach according to geometric distance. Among the approximately 12,000 allocated VSs, most were categorized as flag 10 (71.7%). Flags 10 and 20 were mainly located in upstream and midstream reaches, whereas flags 30 and 40 were mainly located downstream. Approximately 0.8% of VSs showed bias, with considerable elevation differences ($\geq |15|m$) between the mean observed WSE and MERIT digital elevation model. These biased VSs were predominantly observed in narrow rivers at high altitudes.

Following VS allocation using AltiMaP, the median root mean squared error of simulated WSEs compared to satellite altimetry was 7.86 m. The error rate was improved meaningfully (10.6%) than that obtained using a traditional approach, partly due to bias reduction. Thus, allocating VSs to a river network using the proposed AltiMaP framework improved our comparison of WSEs simulated by the global hydrodynamic model to those obtained by satellite altimetry. The AltiMaP source code (https://doi.org/10.5281/zenodo.7597310) (Revel et al.,

2023a) and data (https://doi.org/10.4211/hs.632e550deaea46b080bdae986fd19156) (Revel et al., 2022) are freely accessible online and we anticipate that they will be beneficial to the international hydrological community.

## 1 Introduction

Limited freshwater resources could impede the daily demands of future generations. Monitoring freshwater resources is critical for determining the availability of water for human use. Although continental surface water dynamics can be explored through global-scale hydrodynamic modeling, the effective modeling of freshwater dynamics requires calibration using observed variables such as water surface elevation (WSE), river discharge, and water surface area. Thus, inadequacies of monitoring stream gauges can hinder the performance of hydrodynamic models and fail to accurately represent surface water dynamics (Hannah et al., 2011), such that model evaluation and calibration must depend on remotely sensed data (Meyer Oliveira et al., 2021; Modi et al., 2022; Zhou et al., 2022). Therefore, recent advances in satellite technology have considerably improved our understanding of surface water dynamics.

Satellite altimetry has facilitated direct and reasonably accurate measurements of terrestrial water levels over the past 30 years, with uncertainties ranging from a few centimeters to a few decimeters depending on the environment and altimeter employed. (Cretaux, 2022; Papa et al., 2022). Satellite altimeters determine WSEs by considering differences in the travel time of radar or lasers between the satellite and the water surface. Differences between satellite orbit and altimetry range measurements are used to determine the height of the water surface following dry troposphere, wet troposphere, ionospheric, and/or solid tide correction (Calmant et al., 2008). Several radar altimetry missions have been employed to observe lakes and large rivers, including Topography Experiment (TOPEX)/Poseidon; European Remote Sensing (ERS)-1 and -2; Joint Altimetry Satellite Oceanography Network (Jason)-1, -2, and -3; GEOSTAT Follow On (GFO); Environmental Satellite (ENVISAT); Satellite with ARGOS and ALTIKA (SARAL)-AltiKa; Sentinel-3A, -3B, and -6MF (Calmant et al., 2008; Crétaux et al., 2009, 2011; Santos da Silva et al., 2010; Yang et al., 2022). An updated list of orbit characteristics including temporal resolution, inter-track distance, and frequency for satellite missions that have collected WSE observations is provided in Table 1. In particular, satellite temporal resolution and inter-track distance govern the temporal and spatial resolution of altimetry data. Higher temporal resolution, achieved through frequent passes or shorter revisit times, captures temporal changes with finer granularity, while a smaller inter-track distance provides a higher spatial resolution by offering closely spaced measurements. Consequently, a combination of higher temporal and spatial resolutions in satellite altimetry data enhances the ability to monitor the dynamic processes in the terrestrial surface waters.

Any intersection of a satellite track with a water body is considered a virtual station (VS). The allocation of VSs permits a satellite to retrieve successive water levels at each pass (Santos da Silva et al., 2010). The river width and shape, surrounding topography, and land cover are important factors influencing successful water level retrievals, although no single factor is solely predictive of water level accuracy (Maillard et al., 2015). As a result, radar altimetry retrievals of river surface height depend on the high dielectric constant of water, which causes rivers to reflect more radar radiation than land. It is also challenging to identify exact VS locations due to satellite orbit drift. Therefore, the location of a VS is frequently recorded as the center point of the search area for water level retrieval (Coss et al., 2020; Santos da Silva et al., 2010). To facilitate comparative analyses between satellite observations and numerical simulations, caution must be exercised when transforming the latitude and longitude coordinates of VSs to the river network of the hydrodynamic model.

Satellite altimetry observations have been applied in several large-scale studies to monitor natural water resources in rivers and lakes (e.g., Asadzadeh Jarihani et al., 2013; Birkett et al., 2002; Calmant and Seyler, 2006; Dettmering et al., 2020; Schneider et al., 2017; Xiang et al., 2021), calibrate or validate hydrological/hydrodynamic models (e.g., Elmer et al., 2021; Jiang et al., 2019, 2021; Kittel et al., 2021; Meyer Oliveira et al., 2021; Zhou et al., 2022), and for assimilation into hydrological/hydrodynamic models (e.g., Brêda

et al., 2019; Michailovsky et al., 2013; Paiva et al., 2013; Revel et al., 2023b). However, incorrect VS allocation can lead to the degradation of post-calibration model performance. Thus, the accurate identification of appropriate VS locations within the relevant river reach in the model space is crucial for the comparison of simulation and observation data, as well as for the effective utilization of satellite altimetry in model calibration and validation.

Large-scale hydrodynamic models typically simulate the water dynamics of discretized river segments (i.e., river

grids). The slopes of natural rivers are continuous, whereas elevations are discontinuous among river grids; thus, the digitized VSs can be located between river grids. Physically based hydrodynamic models simulate WSEs with respect to a representative elevation within the river grid which were upscaled from high-resolution hydrography data (i.e., the lowest elevation of high-resolution pixels within the river grid) (Yamazaki et al., 2009, 2011). As a result, the ground elevation of the simulation and observation location can be different, leading to elevation bias

between simulated and observed WSEs. Furthermore, river networks are typically delineated using digital elevation models (DEMs), which suffer from inherent errors (Hawker et al., 2019, 2022; Yamazaki et al., 2017). Therefore, river networks used in large-scale models may contain deviations from the courses of actual rivers (Amatulli et al., 2022; Paz et al., 2006; Yamazaki et al., 2009). To understand the ability of large-scale hydrodynamic models to represent actual WSEs, which is critical for comparing and validating the simulated

WSE, an understanding relative location of VS within the river grid is needed.

Apart from other model limitations such as uncertainty in model parameters, simplified physics, and bias in forcing, the discrepancy in the virtual station location in the river network is a considerable contributor to the bias in simulated water surface elevation when compared to satellite altimetry observations. Large-scale model calibration studies have utilized WSE anomalies for comparison with simulations, where the rough allocation of

VSs in the river proves suitable (e.g., Meyer Oliveira et al., 2021; De Paiva et al., 2013). Conversely, small-scale studies have manually allocated VSs along the river centerline (e.g., Domeneghetti et al., 2021; Jiang et al., 2019, 2021a; Schneider et al., 2017). Calibrations requiring absolute WSE observations, such as calibration of river bottom elevation using rating curves, demand meticulous allocation of virtual stations (VSs) within the river pixels (Zhou et al., 2022). To effectively utilize satellite altimetry observations for supporting large-scale hydrodynamic

model development, a method is required to map representative locations of VSs to relevant river pixels. Moreover, an automated mapping approach becomes essential to facilitate the global-scale model evaluations. Therefore, the development of an automated method for mapping VSs into the river network is paramount to the evaluation of hydrodynamic models on a global scale.We introduce our automated altimetry mapping procedure (AltiMaP), which enable better evaluation of WSEs simulated by large-scale hydrodynamic models using available satellite

altimetry data. AltiMaP reduces the incidence of mismatches between VS locations and actual river locations, which are caused by DEM errors, the use of discrete river grids, and the allocation of VSs to the center of the WSE observation search area. We used pre-processed satellite altimetry data obtained from HydroWeb (https://hydroweb.theia-land.fr, last access: 2 February 2023) to assign VS locations to the high-resolution DEM-

based Multi-Error Removed Improved Terrain Hydrography (MERIT Hydro) flow direction map (Yamazaki et
al., 2019). Simulations were conducted using the Catchment-based Macro-scale Floodplain (CaMa-Flood) global
river hydrodynamic model (Yamazaki et al., 2011) which uses an upscaled river network of MERIT Hydro flow
direction map using Flexible Location of Waterways (FLOW: Yamazaki et al., 2009) algorithm, to evaluate VS
allocation accuracy by comparing satellite altimetry WSE observations with simulation results using AltiMaP and
a traditional VS allocation method.


**Table 1: Satellites altimetry missions which are commonly used for water surface elevation observations. Some characteristics are outlined such as nominal orbit period, temporal resolution, intertrack difference, orbit height, inclination, retracker, agency and data source.**

| Satellite | Norminal Orbit Period | Temporal Resolution (days) | Inter-track distacne at Equater (km) | Orbit Height (km) | Inclination (°) | Retracker | Agency | Data Source |
|---|---|---|---|---|---|---|---|---|
| T/P | 1992-2006 | 10 | 315 | 1336 | 66 | onboard | NASA - CNES | PODAAC |
| ERS-1 | 1991-2000 | 35 | 80 | 785 | 98.52 | ICE-1, ICE-2 | ESA | ESA |
| ERS-2 | 1995-2011 | 35 | 80 | 785 | 98.52 | ICE-1, ICE-2 | ESA | ESA |
| GFO | 1998-2008 | 17 | 165 | 784 | 108 | Ocean | US Navy / NOAA | NOAA |
| ENVISAT | 2002-2012 | 35 | 80 | 800 | 98.55 | ICE-1 | ESA | ESA |
| Jason-1 | 2001-2013 | 10 | 315 | 1336 | 66 | ICE | NASA - CNES | AVISO |
| Jason-2 | 2008-2016* | 10 | 315 | 1336 | 66 | ICE-3 | NASA - CNES - EUMESTAT - NOAA | AVISO |
| Jason-3 | 2016-2022* | 10 | 315 | 1336 | 66 | ICE | NASA - CNES - EUMESTAT - NOAA | AVISO |
| SARAL/Alt iKa | 2013-2016* | 35 | 75 | 800 | 98.5 | ICE-1 | ISRO - CNES | AVISO |
| Sentinel-3A | 2016-Current | 27 | 104 | 814.5 | 98.65 | OCOG | ESA | COPERNICUS |
| Sentinel-3B | 2018-Current | 27 | 52 | 814.5 | 98.65 | OCOG | ESA | COPERNICUS |
| Sentinel-6MF | 2022-Current | 10 | 315 | 1336 | 66 | OCOG | ESA | COPERNICUS |

## 2 Data and Methods

Satellite altimetry data are increasingly used in observing surface water dynamics as their availability has
improved. However, it is essential to develop a framework to deploy altimetry data in the calibration and
validation of surface water dynamics simulations. The AltiMaP algorithm was developed for use with the MERIT
Hydro flow direction map, although it can be applied to other flow direction maps using the "deterministic eight
neighbors" (D8) form, in which the downstream direction is determined by one of the eight neighboring pixels.
The CaMa-Flood model discretizes river networks in terms of irregular-shaped unit-catchments and uses the
elevation of the unit catchment river mouth (i.e., the lowest elevation of the unit catchment) as the riverbank
elevation for that river segment. Therefore, to compare observed WSEs with those simulated by a large-scale

hydrodynamic model such as CaMa-Flood, one can allocate VS location to the MERIT Hydro flow direction map and map it into a coarser-resolution river network.

The accurate allocation of each VS to the MERIT Hydro by AltiMaP involves three main steps: conversion of the VS longitude and latitude to the x- and y-coordinates of a 3″ pixel (~90 m × 90 m at the equator), flagging the VS according to the land cover of the pixel, and allocation of the flagged VS to the nearest river channel on the MERIT Hydro flow direction map. This study introduces the concepts and an overview of the satellite altimetry allocation algorithm; the source code (https://doi.org/10.5281/zenodo.7597310, Revel et al., 2023a) and dataset

prepared for HydroWeb using AltiMaP for use with MERIT Hydro (https://doi.org/10.4211/hs.632e550deaea46b080bdae986fd19156, Revel et al., 2022) are provided.

**2.1 Satellite altimetry data**

Satellite altimetry observes water surface heights by measuring the time it takes for radar/laser pulses to bounce back from smooth surfaces. Although satellite altimetry missions were developed for ocean surface observations,

they have increasingly been applied to observe lakes and rivers (Abdalla et al., 2021; Calmant et al., 2008; Calmant and Seyler, 2006; Yang et al., 2022). Several agencies have already processed their original satellite altimetry data and produced data archives for studying WSEs, including the HydroWeb (Crétaux et al., 2011; Santos da Silva et al., 2010), Hydrosat (Tourian et al., 2016, 2022), Database for Hydrological Time Series of Inland Waters (DAHITTI; Schwatke et al., 2015), Global Reservoirs and Lakes Monitor (G-REALM; Birkett and Beckley, 2010),

Copernicus Global Land Service (CGLS; Calmant et al., 2013; Crétaux et al., 2011), River & Lake (Birkett et al., 2002), Hidrosat (Santos da Silva et al., 2010; da Silva et al., 2012), and Global River Radar Altimetry Time Series (GRRATS; Coss et al., 2020) archives. In this study, we utilized satellite altimetry data obtained from HydroWeb (https://hydroweb.theia-land.fr, last accessed on 2 February 2023), which offered 12523 VSs at the time of data acquisition. For the study, we considered all available VSs from HydroWeb due to its convenient data retrieval

process and global coverage. Initially, we identified all the VSs listed in HydroWeb as potential candidates for inclusion in this research.

**2.2 Hydrography data**

An accurate flow direction map is essential for simulating realistic surface water dynamics at the global scale. The river network used in this study is a 3″ flow direction map derived from the MERIT DEM (Yamazaki et al.,

2017) and water body datasets including the Global 1″ Water Body Map (G1WBM; Yamazaki et al., 2015), Global Surface Water Occurrence (GSWO; Pekel et al., 2016), and OpenStreetMap, which are referred to as MERIT Hydro (Yamazaki et al., 2019). The MERIT Hydro generation involved following steps. Initially a "conditioned DEM" was created by lowering the elevation of water pixels in MERIT DEM based on G1WBM, GSWO, and OpenStreetMap. Subsequently, an initial flow direction was determined based on topographic slope using

"Steepest Slope Method". Some adjustments were made to ensure the flow continuity. Finally, endorheic basins were detected using Global 3″ Water Body Map and Landsat tree density maps (Yamazaki et al., 2019). The MERIT Hydro include an adjusted DEM, river width, height over the nearest drainage, flow accumulation area,

and flow direction data. The 3″ MERIT Hydro was used to determine whether VSs were located on land, river, or ocean pixels. The allocation procedure for the higher-resolution flow direction map is described in Section 2.3.

## 2.3 Allocation of VSs to the MERIT Hydro

VSs must be assigned to river network pixels of the hydrodynamic model for accurate comparison of simulated and observed WSEs. The DEM-based river network can deviate from the cause of the actual river due to errors in DEM and low representability of the coarse-resolution of the river network (Amatulli et al., 2022; Paz et al., 2006; Yamazaki et al., 2009). Moreover, the reported location of the VS provided in HydroWeb can be further away from the actual river because HydroWeb provides the center of the search region, within a range of a few kilometers (e.g., 5 km × 5 km). Therefore, an important step in allocating VSs to large-scale hydrodynamic models is to assign each VS to a river centerline on a higher-resolution flow direction map (e.g., MERIT Hydro, at 3″). A schematic diagram of this allocation process is shown in Figure 1. Initially, the satellite altimetry auxiliary data (e.g., longitude and latitude) for each VS were converted into 3″ pixels. Then we flagged each VS according to the land cover of the initial allocation of the pixel, with 10, 20, 30, and 40 representing river channel, land with the nearest single-channel river, land with the nearest multi-channel river, and ocean pixels, respectively (Figure 1). The secondary flags also defined to represents more special cases as defined supplementary Table S1. Finally, we searched for the centerline of the nearest river according to geometric distance and allocated the VS to that location. VSs initially located on land pixels with the nearest multi-channel rivers were allocated to the nearest largest channel of the multi-channel river (considering the upstream catchment area). The AltiMaP identifies multi-channel river by searching in a direction perpendicular to the specified river considering their downstream connectivity. We assume the observation is from the largest river when there are multiple river (Supplementary

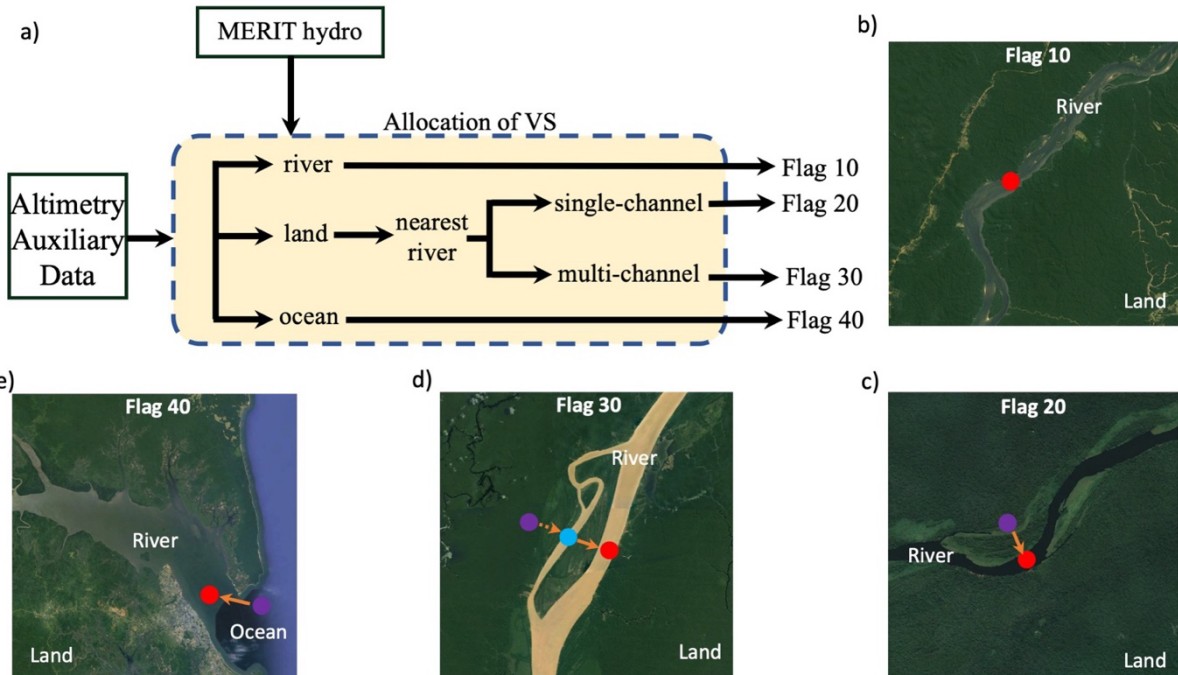

**Figure 1: Schematic diagram of allocating virtual stations (VS) to MERIT-hydro river network. The panels b, c, d, and e present schematics corresponding to Flag 10, Flag 20, Flag 30, and Flag 40, respectively. Red, blue, and purple dots are for final, secondary, and initial locations of VS allocation. (© Google Earth 2022)**

Figure S1) channels near the VS location because backscatter from the narrow river can be highly influenced by non-water features and mostly successful retrievals of WSE can be seen on larger rivers than ~0.8 km. (Birkett et al., 2002).

### 2.4 Filtering biased stations

Even when VSs were aligned perfectly with the river network, simulated WSEs obtained using the river network deviated from satellite altimetry observations. These deviations were caused by errors in the parameters (e.g., riverbank height or river bathymetry [Supplementary Text S1]) and/or the forcings (e.g., surface and subsurface runoff), although satellite altimetry for inland waters can also contain errors (Biancamaria et al., 2017; Frappart et al., 2006; Santos da Silva et al., 2010). The satellite altimetry data should be within a relatively comparable limit with simulated WSE to calibrate or validate the large-scale hydrodynamic models. Since the ground elevations were not recorded at the VS, we compared the mean of the satellite altimetry WSE at the VS with MERIT DEM elevation corresponding to the allocated locations of that particular VSs in the MERIT Hydro flow direction map at 3″-resolution. Then we removed VSs with mean WSEs that were ≥ 15 m higher or lower than the MERIT DEM elevation of the corresponding pixel. These limits were selected in consideration of variation in the flow (Coss et al., 2020) and flood wave height of large rivers (Trigg et al., 2009). We determined that these constraints would be sufficient to include any river surface measurements within a comparable limit, given the elevation data used in this study; however, this threshold can be changed readily to meet user requirements. An example of the application of these restraints for a main Congo channel is provided in Figure 2, in which an unreasonably high VS allocation was removed as biased.

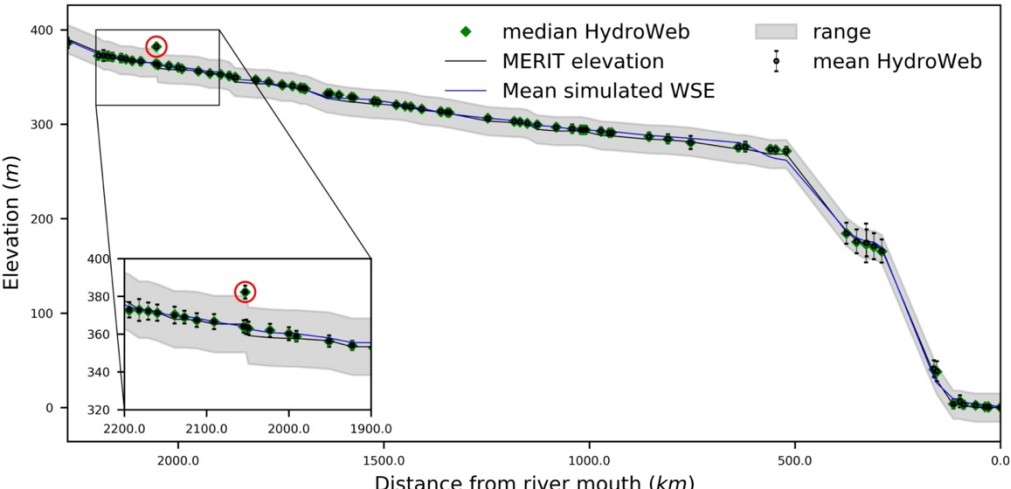

**Figure 2: Example of virtual station (VS) showing unrealistic observations in Congo mainstream. MERIT riverbank elevation; upper and lower limit; and mean simulated WSE using CaMa-Flood hydrodynamic model with VIC BC runoff is shown in grey lines, grey shades, and blue lines, respectively. Black dots and green diamonds indicates the mean and median satellite altimetry height. The standard deviation is shown in black error bars.**

### 2.5 Comparison with simulated WSEs

We used the CaMa-Flood v4.0 model (Yamazaki *et al.*, 2011), which has a spatial resolution of 6′ to evaluate the performance of the AltiMaP VS allocation method. CaMa-Flood determines river hydrodynamics using a local
inertial flow equation (Bates et al., 2010; Yamazaki et al., 2011). The model is forced by runoff (surface and subsurface water flow per unit area) from a land surface model (LSM) to route the water through a river. CaMa-Flood is a physical model that simulates floodplain dynamics and complex hydrodynamics including the hysteresis (Yamazaki et al., 2011, 2012), and flow bifurcation (Yamazaki et al., 2014b). Incorporating accurate DEMs such as MERIT DEM (Yamazaki et al., 2017, 2019) into the CaMa-Flood has enabled it to represent WSE
dynamics more accurately compared to satellite altimetry (Modi et al., 2022). Because CaMa-Flood uses the lowest elevation of the unit-catchment as the elevation of the river segment, and VSs are located where the satellite track crosses the river, which may occur elsewhere within the unit catchment, there may be elevation differences between observed and simulated WSEs (Figure 3). Therefore, evaluating elevation differences between VS locations and unit catchment outlets is important.

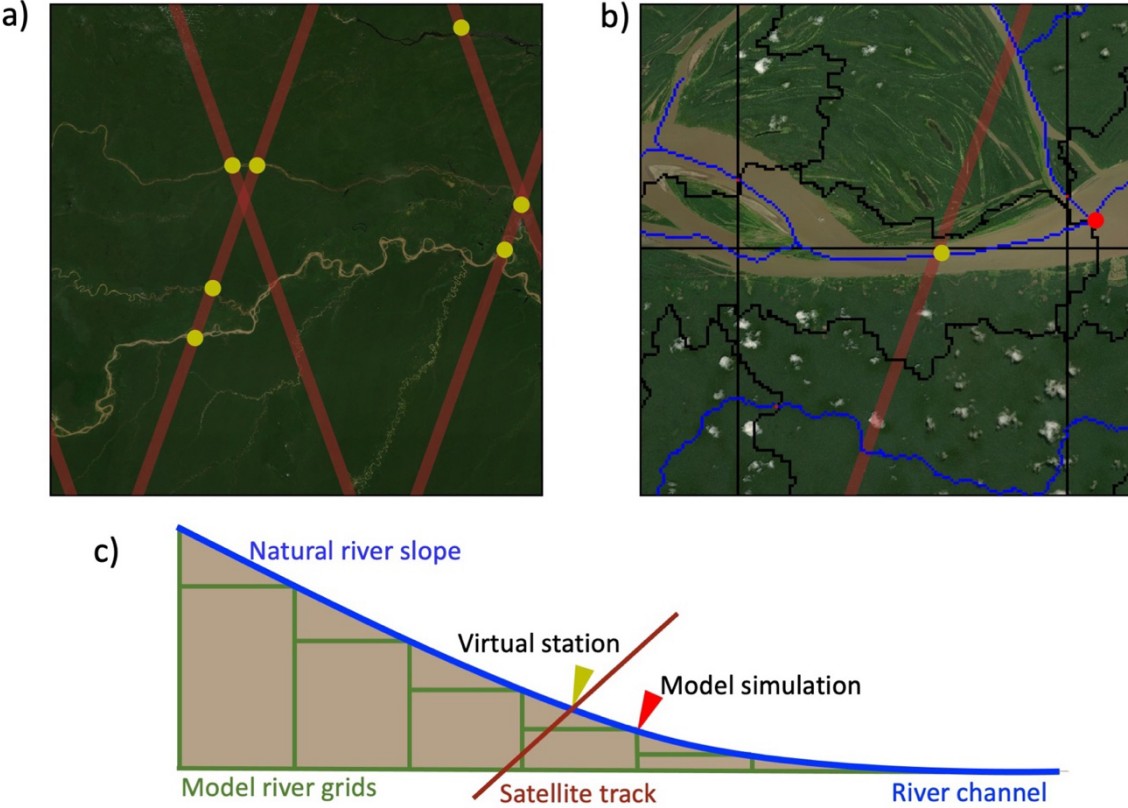

**Figure 3: Representation of Virtual Station (VS) in the river network map for large-scale hydrodynamic model. a) Satellite tracks, b) VS representation in unit-catchment, and c) longitudinal section of the river. Yellow and red color points indicate the VS locations and unit-catchment mouth. The model simulation is corresponding to the unit-catchment mouth corresponding to red point. (Aerials are from Esri, DigitalGlobe, GeoEye, i-cubed, USDA FSA, USGS, AEX, Getmapping, Aerogrid, IGN, IGP, swisstopo, and the GIS User Community)**

We forced the CaMa-Flood hydrodynamic model using the runoff simulated by the Variable Infiltration Capacity (VIC) LSM (Liang et al., 1994) with bias correction (VIC BC) (Lin et al., 2019). The standard model parameters were used in this simulation including parameters such as river bathymetry, river width, and Manning's coefficient. For comparison with WSEs simulated by CaMa-Flood, we mapped VSs to a 6′-resolution global river network after allocating VSs to the MERIT Hydro network at 3″-resolution using AltiMaP, because the CaMa-Flood river map was derived by upscaling the MERIT Hydro flow direction map using FLOW algorithm (Yamazaki et al., 2009). Then we compared the resulting simulated WSEs with observed WSEs mapped onto the river network based on the MERIT Hydro using the AltiMaP algorithm and the ordinary allocation method, i.e., converting longitude and latitude to the CaMa-Flood grid. In this evaluation, our primary objective is to assess the potential improvement brought about by the AltiMaP method when comparing simulated WSE with the ordinary allocation method. For a fair and unbiased evaluation, we employ the same dataset for both observations (i.e., satellite altimetry) and simulations. By doing so, we create a consistent and controlled environment to assess the performance of the AltiMaP method in comparison to the ordinary allocation method. We would like to emphasize that our intention is not to treat the CaMa-Flood simulation results as an absolute reference. Rather, we utilize them as a basis for evaluating the allocation methods concerning satellite altimetry data. Our aim is to investigate whether the AltiMaP method offers any notable advancements in the accuracy of simulated WSEs when compared to satellite-derived measurements.

**2.6 AltiMaP variable identification**

The AltiMaP variables provided for each VS are listed in Table 2; the full dataset is provided in https://doi.org/10.4211/hs.632e550deaea46b080bdae986fd19156 (Revel et al., 2022). The data primarily includes variables related to VS metadata, VS allocation to the MERIT Hydro, and VS mapping to a coarse-resolution river network (e.g., global 6′). The VS metadata consists of the VS ID, name, longitude, latitude, and satellite name. Important parameters for VS allocation that are related to the MERIT Hydro river network can also be calculated for other river network datasets, by flagging and allocating VSs as described in Section 2.3, and then adding 100 to the flag of any VS that is biased (Section 2.4). The distance from a VS mapped to a river centerline to the unit catchment river mouth is an important parameter for understanding differences in water surface dynamics between simulated and satellite altimetry observations. The best and second-best candidate locations for VSs on the MERIT Hydro river centerline (10° × 10° grid) are also reported, along with their geometric distances from the VS location; for single-channel rivers, these data are not available. The river width at each VS location mapped onto the MERIT Hydro river network was calculated using satellite-based water masks and flow direction maps (Yamazaki et al., 2014a). The distance from the VS to the best and second option locations on MERIT Hydro is also included. The coarse-resolution river network variables include the x and y coordinates for the global 6′ map used in the large-scale hydrodynamic model, as well as the elevations of the Earth Gravitational Model 2008 (EGM08) and Earth Gravitational Model 1996 (EGM96).

**Table 2: AltiMaP data description. The data can be divided into three basic categories namely, VS metadata, MERIT Hydro-related, and coarser-resolution river network-related.**

| Variable | Description | Units |
| --- | --- | --- |

| VS metadata | | |
| --- | --- | --- |
| ID | Identification number of VS | - |
| station | VS name | - |
| dataname | dataset name | - |
| lon | longitude | degrees east |
| lat | latitude | degrees north |
| satellite | name of the satellite | - |
| **MERIT Hydro-related** | | |
| flag | allocation flag | - |
| elevation | elevation at VS location on MERIT Hydro | m |
| dist_to_mouth | distance to the unit-catchment mouth | km |
| kx1 | best x-coordinate with respect to the 10° × 10° higher resolution tile | - |
| ky1 | best y-coordinate with respect to the 10° × 10° higher resolution tile | - |
| kx2 | second-best option of x-coordinate with respect to the 10° × 10° high-resolution tile | - |
| ky2 | second-best option of y-coordinate with respect to the 10° × 10° high-resolution tile | - |
| dist1 | distance from the second-best location to the VS | km |
| dist2 | distance from the second-best location to the VS | km |
| rivwth | River width of the allocated location | m |
| **Coarse-resolution river network-related** | | |
| ix | x-coordinate with respect to coarse resolution | - |
| iy | y-coordinate with respect to coarse resolution | - |
| EGM08 | EGM 2008 datum elevation | m |
| EGM96 | EGM 1996 datum elevation | m |

## 3 Results

The AltiMaP dataset produced allocation locations for 12,523 VSs worldwide that are listed in the HydroWeb database. In this section, we discuss the characteristics of VS flags and conditions that can lead to considerable bias in satellite altimetry compared to the MERIT DEM.

**3.1 Allocation of VSs to the river network**

Figure 4a shows the global distribution of flags 10, 20, 30, and 40, which VSs initially located on river channel, land with a single-channel river nearby, land with a multi-channel river nearby, and ocean pixels, respectively. Flag 10 was the most common, accounting for 71.74% of all VSs, followed by flags 20 (26.88%), 30 (1.34%), and 40 (0.04%). Flags 10 and 20 were evenly distributed worldwide. Mostly, large rivers such as Amazon, Congo, Nile, Ob, etc. consist of flags 10 or 20 which indicate the low inconsistencies between VS locations and the river network. Flag 40 is distributed near the ocean in Congo River, Santee River in United States, Lumi Semanit River in Albania, Mahavavy River in Madagascar, and Luni River in India. In addition, flag 30 can be seen mostly in mid-streams where multi-channel rivers exist. Hence, different flags shows different geographical characteristics.

The log probability distributions of upstream catchment areas for different flag values are also shown in Figure 4b. The median upstream catchment areas were $2.73 \times 10^4$, $9.95 \times 10^3$, $2.16 \times 10^4$, and $3.95 \times 10^4$ km$^2$ for flags 10, 20, 30, and 40, respectively. Flag 40 represented the largest median upstream catchment area because those are closer to the ocean and have large upstream catchment area. The distribution of flag 40 was strongly right skewed, influenced by the larger upstream catchment area of downstream Congo River. Flag 20 had the smallest median upstream catchment area, which indicates that most flag 20 VSs were in upstream reaches.

Figure 4c depicts the probability distribution of riverbank elevation for each flag. Lines represent the probability distributions of elevation for flags 10 to 30, with median values of 112.9m, 147.0m, and 141.2m for flag 10, flag

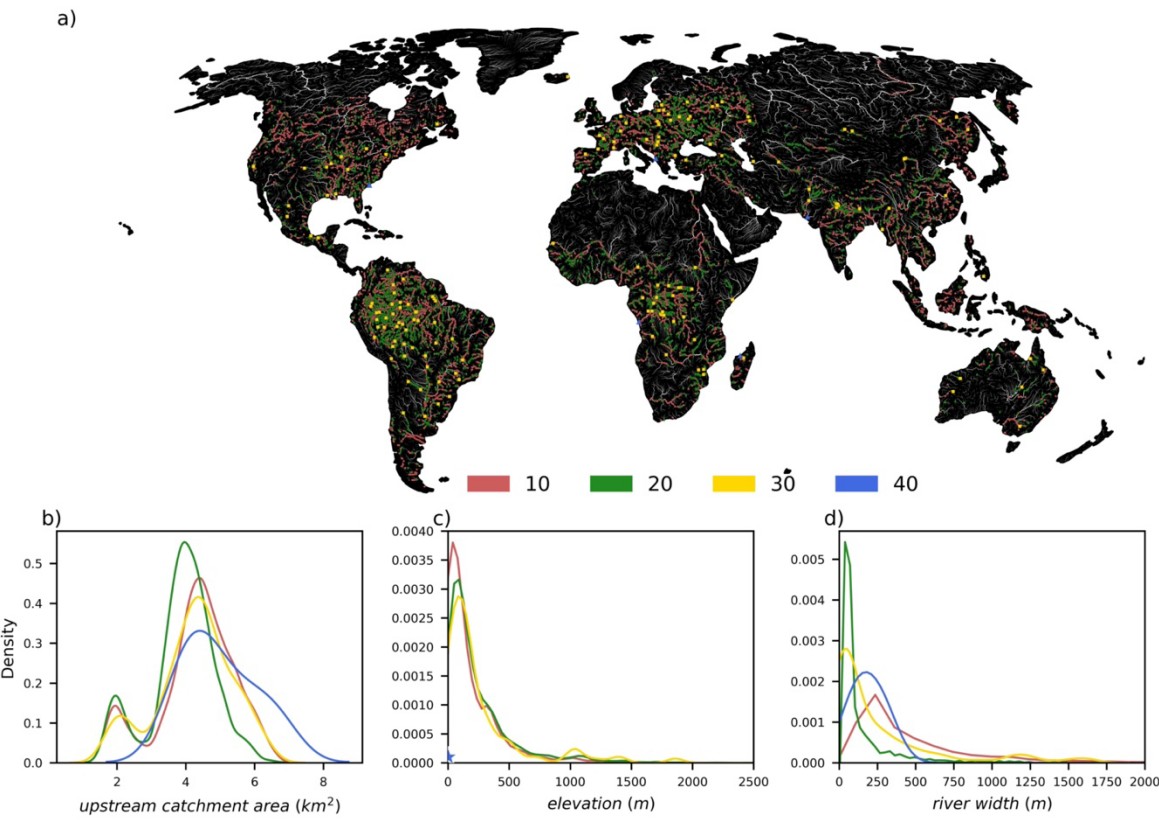

**Figure 4: Global map of allocation flags. Panel at lower left corner shows probability distribution of the upstream catchment area in log scale for different flags. Flags 10, 20, 30, and 40 are indicated by light blue, medium blue, dark blue, and red colors, respectively.**

20, and flag 30, respectively. Notably, flag 40 was not visible in Figure 4c due to its very low elevation, with a median of 0.0m (mean=0.54m and std=1.21m). Flags 10 to 30 were distributed from mean sea level to 4790.0m, and there was no significant difference in elevation observed among flags 10 to 30.

The river width distribution for each flag is demonstrated in Figure 4d. Flag 20 exhibited the smallest median river width at 41.4m, with a relatively low standard deviation of 193.3m. On the other hand, Flag 40 displayed the largest median river width of 224.0m, but its variation was substantial (std=1336.6m) due to the wider Congo downstream, which measures around 3170.0m. Flag 10 showed a median river width value of 222.0m, comparable to Flag 40, but with a lower variation (std=683.6m). Meanwhile, Flag 30 exhibited a median river width of 77.7m,

falling between the median river widths of Flag 10 and Flag 40. The large variation in river width observed for Flag 10 was due to its widespread distribution across the rivers, while the substantial variation of Flag 40 was influenced by the VSs' location in the Congo River.

**3.2 Biased VSs**

Figure 5 shows the spatial heterogeneity of biased VSs, their distribution of upstream catchment areas in log scale,

variation in their elevations, and a histogram of river widths at VS locations, calculated using MERIT Hydro. Biased VSs accounted for 2.6% of all VSs, and were distributed worldwide, with no distinct spatial pattern. A large number of them were allocated to large river basins such as the Amazon, Congo, and Mekong basins. Most

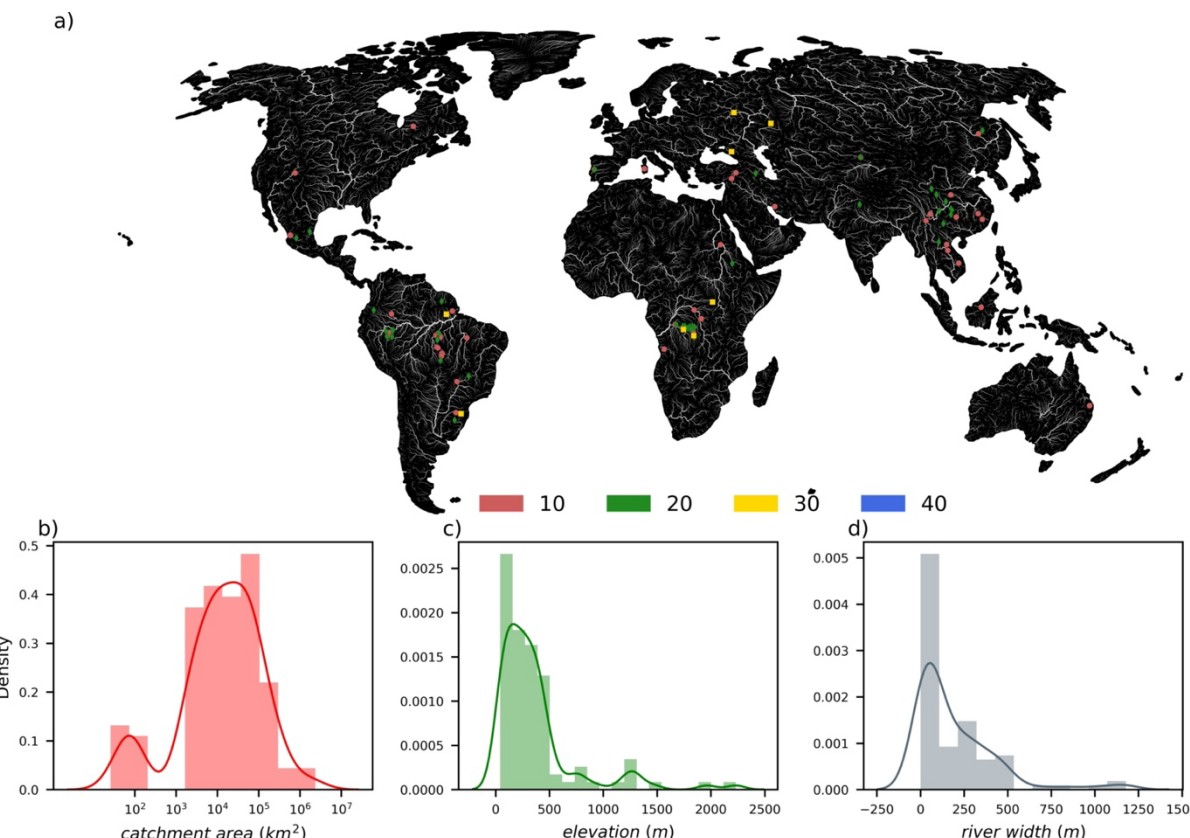

**Figure 5: a) Global distribution, b) histogram of catchment area (km²), c) histogram of elevation (m), and d) histogram of river width (m) of biased VSs. Light blue circles, medium blue diamonds, dark blue squares, and red triangles for flags 10, 20, 30, and 40, respectively in panel a.**

were flagged 20, which was the second most common allocation flag. Many were detected in the Amazon basin of South America. The median upstream catchment area of biased VSs was $2.98 \times 10^4$ km$^2$, their median elevation was 199.6 m, and the median river width was 87.5 m, with most values ranging from 0 to 500 m. Thus, most biased VSs were detected in narrow rivers at high altitudes.

VSs with biased WSEs were generally found in narrow, high-elevation river reaches, although some were found in rivers such as the main Congo channel. Most biased VSs had WSEs that exceeded the MERIT Hydro feasible elevation range. Large biases can be caused by off-nadir measurement of nearby water bodies (Maillard et al., 2015), deviations of the MERIT Hydro river network from actual river (Amatulli et al., 2022), and DEM errors such as vegetation bias (Yamazaki et al., 2017). Further study is needed to fully understand the causes of errors in river WSEs obtained by satellite altimetry which is beyond the scope of this data description paper.

## 4 Discussion

### 4.1 Effect of discrete river reaches and DEM errors.

As the distance from the VS location to the unit catchment mouth increased, the median RMSE of simulated WSE increased (Figure 6), mainly due to the difference in elevation between these points. Thus, large errors may be associated with simulated WSE when the VS is located far from the unit catchment mouth. Similarly, the median RMSE of simulated WSE increased slightly as the slope within the unit-catchment increased until slope < 200 m/km, with larger slopes (> 200 m/km) showing an increase in median RMSE from 2 to 4 m. This variation may have been caused by the non-uniformity of slopes within unit catchments of the CaMa-Flood model; however, it was well within the range of variation within unit-catchment slope bins, which reached up to 8 m.

One of the main reasons for elevation bias between the satellite altimetry and model simulations is elevation differences between the VS locations and the base elevation of the model. This type of bias can be eliminated using the VS location as the unit-catchment mouth. However, this approach is challenging because unit-

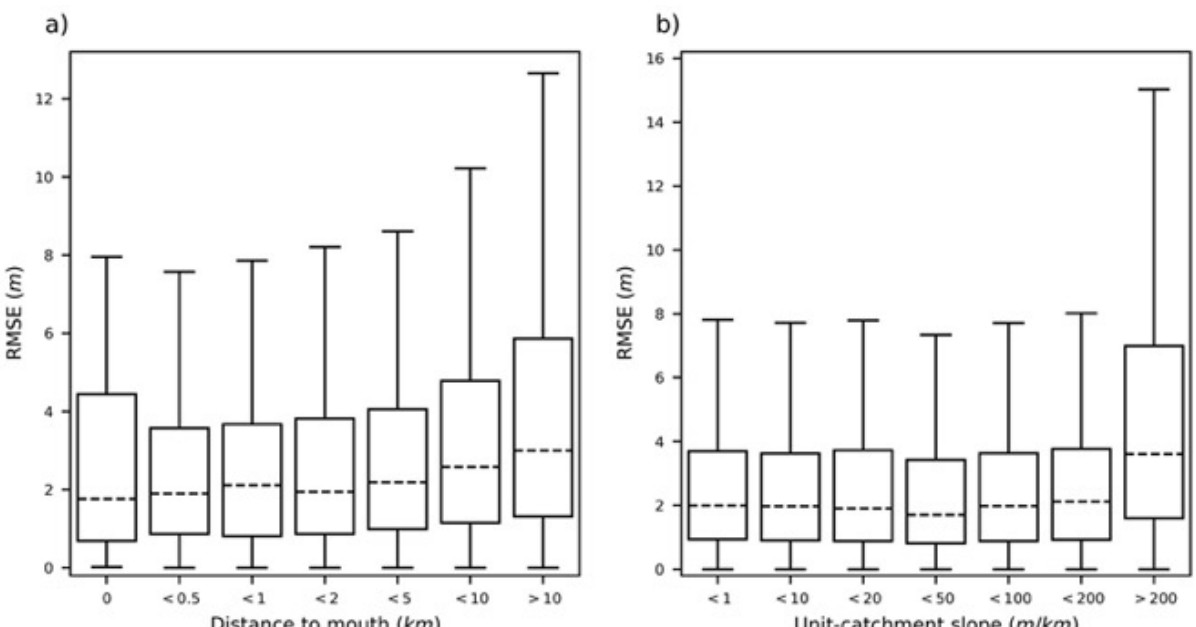

**Figure 6: Boxplot of the root mean square error (RMSE) against a) the distance to the unit-catchment mouth and b) unit-catchment slope.**

catchments size may be very small when several VSs located close to each other, which may lead to computational instability in CaMa-Flood model because it is optimized for unit-catchments of equal size. In addition, changing unit-catchment sizes can reduce the computational efficiency of the model drastically, which is critical for global-scale hydrodynamic models such as CaMa-Flood. Therefore, we did not consider updating the river network to use the VS locations as unit-catchment mouths in AltiMaP.

The allocation of VSs to a river network is highly dependent on the DEM used to delineate the river network (Schumann and Bates, 2018). Most freely available global-scale DEMs have large vertical errors that are accentuated over complex topography; these are unable to resolve microtopographic variation in relatively flat terrain (Chu and Lindenschmidt, 2017; Gallien et al., 2011). Although global-scale DEMs such as the Advanced Spaceborne Thermal Emission and Reflection Radiometer (ASTER) or Shuttle Radar Topography Mission

(SRTM) exhibit non-negligible height errors, recent studies have attempted to eliminate these errors (e.g., Hawker et al., 2022; Rizzoli et al., 2017; Yamazaki et al., 2017). In this study, we used the MERIT DEM, which is a highly accurate global-scale DEM that is freely available (Hawker et al., 2019). Thus, AltiMaP can be applied to river networks delineated using any accurate global DEM.

**4.2 VS allocation to MERIT Hydro**

Mapping the VSs to MERIT Hydro, a high-resolution global river network is a crucial step in leveraging their potential for hydrological modeling. There are several compelling reasons for mapping the VSs to MERIT Hydro, which is a high-resolution global river network at 3″. Firstly, the mapping process can be easily adapted to various resolution river networks of the CaMa-Flood hydrodynamic model, such as 0.25° or 0.1°. This flexibility allows for the integration of VSs into a range of hydrological models, depending on the desired level of detail and

accuracy. Secondly, the relative location of the VSs within the CaMa-Flood unit-catchment can be determined, which is essential for the calculation of important parameters such as elevation difference and distance to the unit catchment mouth (dist_to_mouth). These parameters are critical for evaluating and understanding the dynamics of water in a river network. Finally, the ability to allocate VS to any river network with a similar topology is

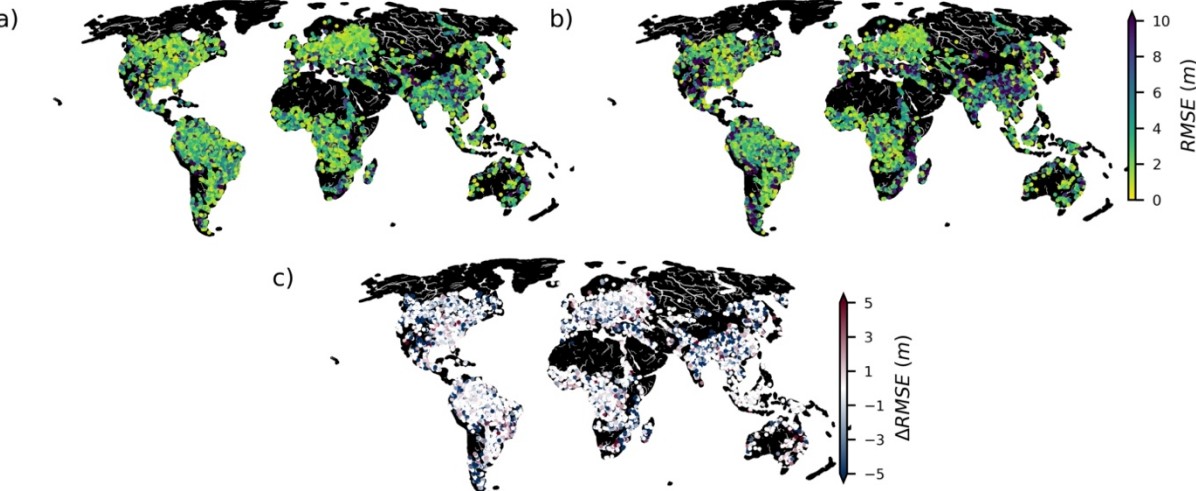

**Figure 7: Global map of root mean square error ($RMSE$) for a) AltiMaP and b) ordinary method; and c) RMSE difference ($\Delta RMSE$) between expert and ordinary methods.**

demonstrated by mapping the VSs to MERIT Hydro, which is having D8 connectivity. Therefore, it is crucial to
assign the VSs to the base map of the river network in the hydrodynamic model to enhance the evaluation of the models and to identify the causes for the discrepancies between the model and observations.

RMSEs were calculated for WSEs simulated by CaMa-Flood and forced by VIC BC runoff (Lin et al., 2019). Both simulations and observations were converted to the same geoid before calculating RMSE (i.e., EGM96). The spatial distributions of WSE RMSEs for VS allocations obtained using AltiMaP and the traditional method
of allocating VSs to the CaMa-Flood grid are shown in Figure 7. Traditional VS allocation was performed using directly converting longitude and latitude information to coarse-resolution (i.e., 0.1°) grids. At the global scale, RMSEs were generally similar between both VS allocation methods. However, the satellite altimetry was better represented by AltiMaP for 17.52% of VSs (negative ΔRMSE) and by the traditional method for only 12.85% of VSs (positive ΔRMSE) The lower ΔRMSE of ordinary method may be due to the fact allocation to a nearby grid
by ordinary method compensate for the errors in the model such as river bathymetry error (Modi et al., 2022).

The AltiMaP and traditional VS allocation methods had median RMSEs of 7.86 and 8.70 m, respectively (Figure 8). The inter-quartile range was larger for the traditional method. Thus, AltiMaP reduced the RMSE by 10.6% through more accurate VS allocation to the river network map. RMSE was reduced by AltiMaP for all flags, with the largest reduction observed in flag 30 (Table 3) due to VS allocation to more appropriate segments of multi-
channel rivers, followed by flags 20, 10. But accuracy was slightly degraded in flag 40, in which VSs were allocated inward from the ocean. The remaining error may be attributed to elevation differences between VS locations and simulated WSE locations (Figure 3) and limitations of the hydrodynamic model.

A flag-wise comparison revealed that errors associated with the allocation method varied among flags in the AltiMaP results. Almost all AltiMaP flags had lower RMSE than those produced by the traditional method. This
difference was due to the irregular shape of unit-catchments in the CaMa-Flood hydrodynamic model. In long, narrow unit catchments, slight deviations in VS location could lead to the misidentification of adjacent unit catchments as target grid. Thus, simulated WSEs deviated by 1–15 m, depending on the slope and river path (e.g., straight vs. meandering river). These results highlight the importance of implementing specialized procedures such as AltiMaP to locate optimal river grid matches for each VS prior to WSE comparisons.

**4.3 Advantage of mapping VSs**

Because we used river network-related variables in the AltiMaP VS allocation algorithm, we were able to calculate distances and elevation differences between each VS and the unit-catchment river mouth. These parameters are particularly important for comparing WSEs simulated by coarse-resolution, large-scale river routing models such as CaMa-Flood, which are based on discretized river reaches with a representative elevation for each pixel.
Minimizing the distance and elevation difference between the VS and unit-catchment river mouth is critical for improving the accuracy of WSE simulations. Thus, this elevation difference may be used as a proxy to interpret bias between simulated and observed WSEs (Fassoni-Andrade et al., 2021). Satellite altimetry data are also extremely useful for evaluating and calibrating hydrodynamic models (e..g., Zhou et al., 2022) and correcting variables through data assimilation (e.g., Revel et al., 2023b), which requires correct VS allocation to a river
network map. The river bathymetry parameter can be calibrated using rating curve method developed using

satellite altimetry and in-situ river discharge data (Zhou et al., 2022). Furthermore, the model can be evaluated using multi-variables (i.e., river discharge, WSE, and inundation extent) (Modi et al., 2022).

The flags used in AltiMaP to classify VSs provide a unique opportunity for users to identify the VS allocation methods used to evaluate hydrodynamic model outputs. Notably, simulated WSEs in first- and second-candidate river pixels for VSs that were initially allocated to multi-channel rivers (flag 30) can be used to select optimal VS locations along the river network. Most VSs flagged 10 were located in upstream reaches, whereas those flagged 30 and 40 were initially located in multi-channel rivers and oceans (which are most in need of relocation) and were allocated to downstream reaches. It is important to correctly allocate VSs initially located on multi-channel rivers because river networks based on the MERIT Hydro separate each channel of a multi-channel river into different unit-catchments. Thus, discrepancies in the allocation of VSs located on smaller channels can mistakenly alter the WSE dynamics of the simulation, such that the allocation flags are important indicators of VS usage in the context of hydrodynamic modeling.

**4.4 Limitations and Future Perspectives**

Even though AltiMaP is suitable in mapping the VSs into the given river network with D8 connection, the method is not capable of identifying non-nadir observations (such as floodplain lakes near the river channel). One of the major problem in the conventional altimeters in low-resolution mode (LRM) such as ENVISAT was correcting the observations from the non-nadir view was treated as nadir observations (Calmant et al., 2008; Frappart et al., 2006; da Silva et al., 2012). The dual antenna configuration of the CryoSat-2 allows precise position of reflecting point in the radar footprint and solve the signal location along-track and across-track directions (Cretaux, 2022). Moreover, ICESat-1/2 data can also be a great source of importance over terrestrial waters, but the longer revisit time limit the applications in hydrology. Satellites such as CroySat-2 and ICESat-2 provide an addition challenge in using them in river monitoring. CryoSat-2 with its' drifting orbit ~7.5km makes it challenging to define VSs as in repeat orbits (Schneider et al., 2017). With the complex ground track configuration of ICESat-2 makes it complex to use in river monitoring because the assigning method would differ depend on the satellite track orientation with respect to the river centerline (Scherer et al., 2023). However, with slight modification to the AltiMaP, we would be able to map such data into the MERIT Hydro (Supplementary Figure S2).

**5 Data availability**

Data produced by AltiMaP were published in https://doi.org/10.4211/hs.632e550deaea46b080bdae986fd19156 (Revel et al., 2022). MERIT Hydro river network data are freely available (http://hydro.iis.u-tokyo.ac.jp/~yamadai/MERIT_Hydro/) under a Creative Commons license (CC-BY-NC 4.0).

**6 Code availability**

The AltiMaP algorithm was published in https://doi.org/10.5281/zenodo.7597310 (Revel et al., 2023a) and is available for noncommercial use. The CaMa-Flood source codes are also available (https://github.com/global-hydrodynamics/CaMa-Flood_v4) under the Apache 2.0 license.

**Table 3: Median statistics of the error of simulated WSE using CaMa-Flood hydrodynamic model. RMSE (root mean squared error), Bias, and CC (correlation coefficient) were presented. The simulated WSE is compared with HydroWeb satellite altimetry data where the VS s were allocated using AltiMaP or the ordinary allocation method.**

|         | AltiMaP |       |      | Ordinary |       |      |
|---------|---------|-------|------|----------|-------|------|
|         | RMSE    | bias  | CC   | RMSE     | bias  | CC   |
| All     | 2.68    | -0.01 | 0.67 | 2.98     | -0.99 | 0.67 |
| Flag 10 | 2.65    | -0.43 | 0.67 | 2.94     | -1.87 | 0.68 |
| Flag 20 | 2.71    | -0.17 | 0.66 | 3.06     | -2.46 | 0.66 |
| Flag 30 | 2.72    | -0.60 | 0.64 | 2.85     | -1.97 | 0.61 |
| Flag 40 | 0.85    | -0.37 | 0.02 | 0.94     | -0.30 | 0.02 |

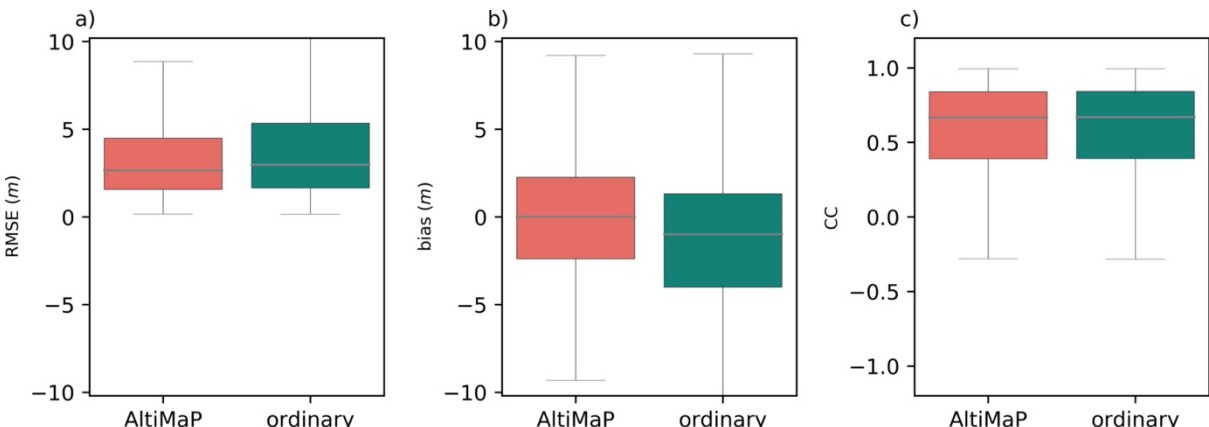

**Figure 8: Distribution of root mean square error (*RMSE*), bias, and correlation coefficient (CC) for AltiMaP and ordinary VS allocation methods in panels a, b, and d, respectively.**

## 7 Summary

We introduce AltiMaP, an effective methodology for comparing satellite altimetry WSE observations with WSEs simulated by large-scale hydrodynamic models such as CaMa-Flood. The procedure involves allocating each VS to a suitable high-resolution (3″) pixel, flagging the pixel according to land cover, and filtering out biased VSs according to the local MERIT DEM elevation. The main objective of this study was to improve the accuracy of allocation to a river network for a useful comparison of simulated and observed WSEs, among other applications.

We compared WSEs simulated by the CaMa-Food hydrodynamic model based on VIC BC runoff to satellite altimetry WSEs based on VS allocation to the MERIT Hydro river network using AltiMaP.

After mapping the flagged VSs to a 6' river network, biased VSs with values above or below the feasible MERIT Hydro elevation range were filtered out. Most VSs were located on single-channel rivers; VSs initially located on land were distributed worldwide. VSs initially located on multi-channel rivers and oceans were allocated to downstream reaches of large rivers such as the Amazon, Congo, and Mekong Rivers. Biased VSs, incompatible with the river network elevation profile, were mainly found in narrow rivers at high elevations, likely because

most altimeters are designed to observe ocean topography. Such VS biases are mainly caused by off-nadir measurements, DEM errors, or errors in the geolocation of river networks.

We also allocated VSs to a coarse-resolution CaMa-Flood river network for comparison with the simulated results. AltiMaP VS allocation represented the satellite altimetry more accurately than a traditional method, reducing the RMSE associated with the simulated WSEs by approximately 10%, representing a difference of approximately 2 m in multi-channel rivers. AltiMaP can be applied to any currently available processed satellite altimetry datasets (e.g., DAHITTI, Hydrosat, and CGLS) and any river network with simple land cover definitions (e.g., river, land, and ocean). We anticipate that the algorithm will contribute to the evaluation and/or calibration of hydrodynamic models using satellite altimetry and the acquisition of accurate hydrodynamic model output through satellite altimetry assimilation.

8 **Author contribution:** MR developed and finalized processing algorithms, performed the exploration of the methods, and finalized the manuscript. MR and DY designed the experiments. XZ and PM provided the data for comparison. JC and SC provided the expertise on deriving the satellite altimetry for rivers. All co-authors involved in revising and editing the manuscript.

9 **Competing interests:** The authors declare that they have no conflict of interest.

## 10 Acknowledgments

This work was supported by the Japan Society for the Promotion of Science (JSPS) under KIBAN-S Grant No.21H05002, JSPS KIBAN-B 20H02251, and JSPS start-up 20K22428. We extend our gratitude to the Topic editor James Thornton, four anonymous referees, and João Paulo Brêda for their valuable comments and suggestions.

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
