# Peer review of "AltiMaP: Altimetry Mapping Procedure for Hydrography Data"

_Earth System Science Data, 2022_

## Author Comment (AC1)

**Anonymous Referee #1 - Comment on essd-2022-438**

We would like to express our gratitude to Referee #1 for insightful comments and suggestion. We have carefully reviewed your comments and have made necessary updates to our manuscript. We provided point-to-point response to the referee comments shown in blue whereas the revision made to the main text in shown *blue Italics*.

Kind regards

I have reviewed the paper "AltiMaP: Altimetry Mapping Procedure for Hydrography Data" by Revel et al. (2023), which introduces a method for allocating virtual stations (VSs) from satellite altimetry data to a river network. This method can improve the accuracy of comparing simulated water surface elevations (WSEs) using satellite altimetry data with those obtained from hydrodynamic models. I find this paper to be well written and recommend it for publication after addressing the following comments and suggestions.

We are thankful to Referee #1 for agreeing to review our manuscript. We appreciate the positive feedbacks from Referee #1.

1)      Line 14: The authors have used numbers 10, 20, 30, and 40 to represent different flags, but it would be helpful if they could provide a more detailed explanation of their reasoning. Why not 1, 2, 3, and 4.

We appreciate the comment by Referee #1. We used flags as 10, 20, 30, and 40 for represent the different altimetry mapping procedures. We used those number rather than 1, 2, 3, and 4 to give some flexibility for adding some subdivision inside each flag. For example, we have given the following table for further assign the altimetry mapping.

*Table S1: Secondary Flags used in the AltiMaP. Here large and small river are with respective to each river section. The upstream catchment area was used to define the small and large rivers.*

| Main Flags | Secondry Flag | Description |
|---|---|---|
| | 10 | VS was found on the river centerline |
| 10 | 11 | VS was found on the river channel but not in the centerline and assigned to the nearest centerline |
| | 12 | VS was found in the unit-catchment mouth |
| 20 | 20 | VS was found in the ground and assinged to the nearest single channel centerline |
| | 21 | VS was found in the ground near large river channel in in mult-channel river and assinged to the larger river centerline |
| 30 | 30 | VS was found in the ground near small river channel in mult-channel river and assinged to the large river centerline |

| 31 | | VS was found in bifuricating channel and assinged to the large river centerline |
|---|---|---|
| 40 | 40 | VS was found in the ocean and assinged to nearest river channel |

For simplicity we have given only the main flags in the manuscript. We will attach the above table as a supplementary.

2) Line 37: The authors should provide statistical measures such as mean absolute error or any to define what they consider to be reasonable accuracy. Please avoid using qualitative words.

Thanking the referee #1, we revised the text as follows:

*"Satellite altimetry has facilitated direct and reasonably accurate measurements of terrestrial water levels over the past 30 years, with uncertainties ranging from a few centimeters to a few decimeters depending on the environment and altimeter employed. (Cretaux, 2022; Papa et al., 2022)."*

3) Lines 42-45: To improve the readability of the paper, it would be helpful if the authors provide separate references for each different radar altimetry mission instead of listing all the references together. This will make it easier for readers to identify and access the specific sources of information relevant to each mission.

The references shown here is the studies who used the mention satellites for observing lakes and rivers. They have used multiple satellite in each respective study. Hence, it may be better to provid them as common citations. Moreover, we provided the details about the retrackers used, agency, data sources in addition to previous Table 1. Updated Table 1 is attached below.

*"Table 1: Satellites altimetry missions which are commonly used for water surface elevation observations. Some characteristics are outlined such as nominal orbit period, temporal resolution, intertrack difference, orbit height, inclination, retracker, agency and data source."*

| Satellite | Norminal Orbit Period | Temporal Resolution (days) | Inter-track distacne at Equater (km) | Orbit Height (km) | Inclination (°) | Retracker | Agency | Data Source |
|---|---|---|---|---|---|---|---|---|
| T/P | 1992-2006 | 10 | 315 | 1336 | 66 | onboard | NASA - CNES | PODAAC |
| ERS-1 | 1991-2000 | 35 | 80 | 785 | 98.52 | ICE-1, ICE-2 | ESA | ESA |
| ERS-2 | 1995-2011 | 35 | 80 | 785 | 98.52 | ICE-1, ICE-2 | ESA | ESA |
| GFO | 1998-2008 | 17 | 165 | 784 | 108 | Ocean | US Navy / NOAA | NOAA |
| ENVISAT | 2002-2012 | 35 | 80 | 800 | 98.55 | ICE-1 | ESA | ESA |
| Jason-1 | 2001-2013 | 10 | 315 | 1336 | 66 | ICE | NASA - CNES | AVISO |
| Jason-2 | 2008-2016* | 10 | 315 | 1336 | 66 | ICE-3 | NASA - CNES - EUMESTAT - NOAA | AVISO |

| | | | | | | | | |
|---|---|---|---|---|---|---|---|---|
| Jason-3 | 2016-2022* | 10 | 315 | 1336 | 66 | ICE | NASA - CNES - EUMESTAT - NOAA | AVISO |
| SARAL/Alti Ka | 2013-2016* | 35 | 75 | 800 | 98.5 | ICE-1 | ISRO - CNES | AVISO |
| Sentinel-3A | 2016-Current | 27 | 104 | 814.5 | 98.65 | OCOG | ESA | COPERN ICUS |
| Sentinel-3B | 2018-Current | 27 | 52 | 814.5 | 98.65 | OCOG | ESA | COPERN ICUS |
| Sentinel-6MF | 2022-Current | 10 | 315 | 1336 | 66 | OCOG | ESA | COPERN ICUS |

4) Table 1: The authors could add an additional column discussing how these different data sources differ from each other in their data generation algorithms.

We would like to thank the referee #1 for the valuable comments. We added few columns to the previous Table 1 as explained above. We added inclination angle, retracker used, agency, and data source. We hope this additional column would give readers information about data accessibility and the differences in the data generation algorithms.

5) Line 46: The authors should provide a brief explanation of how the temporal resolution and inter-track distance of satellite altimetry data affect the temporal and spatial resolution of the data.

We would like to express our gratitude to referee #1 for the valuable suggestions. The frequency of revisits in a repeating satellite orbit determines its temporal resolution. A shorter revisit time means more frequent observations for a single location. Additionally, a smaller inter-track distance results in closer observation locations, with VSs being situated nearer to one another.

We have revised and added the following text to the Introduction section:

*"Higher temporal resolution, achieved through frequent passes or shorter revisit times, captures temporal changes with finer granularity, while a smaller inter-track distance provides a higher spatial resolution by offering closely spaced measurements. Consequently, a combination of higher temporal and spatial resolutions in satellite altimetry data enhances the ability to monitor the dynamic processes in the terrestrial surface waters."*

6) Lines 70-77: The authors should provide a more detailed discussion of existing studies that have attempted to accurately locate VSs, rather than only discussing the problem. By doing so, they can highlight the research gap that their work aims to address and demonstrate how their method contributes to the current state of research on this topic. Improving the research gap in detail will help readers better understand the significance of the authors' contribution and appreciate the originality of their approach.

We would like to thank referee #1 for the suggestions. To our knowledge, this is the first attempt to systematically allocate satellite altimetry locations to the global river network. A few studies have used satellite altimetry for calibration and validation of model outputs but most of them have either compared WSE anomalies or used VSs near the simulation locations (e.g., Meyer Oliveira et al., 2021; De Paiva et al., 2013). Some others compared satellite altimetry with WSE simulated

by the small-scale models (e.g., Domeneghetti et al., 2021; Jiang et al., 2019, 2021). In addition, Schneider et al., (2017) projected the CryoSat-2 observed into the river centerline of Brahmaputra river. In such instances the VS may allocated manually to the river centerline. But in the cases where absolute WSE is needed on large-scale calibration approaches such as in Zhou et al., (2022), an automated robust allocation method is essential. Therefore, we developed an automated mapping method for VSs for large-scale hydrodynamic models.

We have added a paragraph to Introduction Section highlighting the importance of our study as follows (new additions were shown in *purple*) considering the referee #1's suggestions:

*"Apart from other model limitations such as uncertainty in model parameters, simplified physics, and bias in forcing, the discrepancy in the virtual station location in the river network is a considerable contributor to the bias in simulated water surface elevation when compared to satellite altimetry observations. Large-scale model calibration studies have utilized WSE anomalies for comparison with simulations, where the rough allocation of VSs in the river proves suitable (e.g., Meyer Oliveira et al., 2021; De Paiva et al., 2013). Conversely, small-scale studies have manually allocated VSs along the river centerline (e.g., Domeneghetti et al., 2021; Jiang et al., 2019; Schneider et al., 2017). Calibrations requiring absolute WSE observations, such as calibration of river bottom elevation using rating curves, demand meticulous allocation of virtual stations (VSs) within the river pixels (Zhou et al., 2022). To effectively utilize satellite altimetry observations for supporting large-scale hydrodynamic model development, a method is required to map representative locations of VSs to relevant river pixels. Moreover, an automated mapping approach becomes essential to facilitate the global-scale model evaluations. Therefore, the development of an automated method for mapping VSs into the river network is paramount to the evaluation of hydrodynamic models on a global scale."*

7) Line 117: The authors should provide a more detailed explanation of their data selection criteria, such as period and temporal resolution, and explain why they chose to use satellite altimetry data from HydroWeb instead of other sources.

Thank you, for the selection of VSs, we simply select all the VSs listed in the HydroWeb as potential VSs in this study. However, we compared the satellite altimetry observations with the CaMa-Flood simulated WSE from 2002 to 2019 considering the data availability. The methods we presented here can be applied to any pre-processed data set such as HydroWeb, DAHITTI, etc. However, we used HydroWeb in this study because its' global availability and easy access.

We have revised the following text considering the referee #1's suggestions. Revised or added text have been shown in *purple*:

*"2.1 Satellite altimetry data*

*Satellite altimetry observes water surface heights by measuring the time it takes for radar/laser pulses to bounce back from smooth surfaces. Although satellite altimetry missions were developed for ocean surface observations, they have increasingly been applied to observe lakes and rivers (Abdalla et al., 2021; Calmant et al., 2008; Calmant and Seyler, 2006; Yang et al., 2022). Several agencies have already processed their original satellite altimetry data and produced data archives*

*for studying WSEs, including the HydroWeb (Crétaux et al., 2011; Santos da Silva et al., 2010), Hydrosat (Tourian et al., 2016, 2022), Database for Hydrological Time Series of Inland Waters (DAHITTI; Schwatke et al., 2015), Global Reservoirs and Lakes Monitor (G-REALM; Birkett and Beckley, 2010), Copernicus Global Land Service (CGLS; Calmant et al., 2013; Crétaux et al., 2011), River & Lake (Birkett et al., 2002), Hidrosat (Santos da Silva et al., 2010; da Silva et al., 2012), and Global River Radar Altimetry Time Series (GRRATS; Coss et al., 2020) archives. In this study, we utilized satellite altimetry data obtained from HydroWeb (https://hydroweb.theia-land.fr, last accessed on 2 February 2023), which offered 12523 VSs at the time of data acquisition. For the study, we considered all available VSs from HydroWeb due to its convenient data retrieval process and global coverage. Initially, we identified all the VSs listed in HydroWeb as potential candidates for inclusion in this research."*

8) Lines 118-119: The authors could improve the readability by moving the discussion about identifying and removing biased VSs to Section 2.4 and providing a more detailed explanation of their criteria for identifying VSs.

We apricate the comment from the referee #1. We moved the explanation about filtering biased VSs to the Section 2.4.

9) Line 120: The authors could include a brief discussion of each data source used in the study and how the data were derived.

We appreciate the suggestion and add some information about MERIT Hydro to Section 2.2. The added text is highlighted in purple.

*"An accurate flow direction map is essential for simulating realistic surface water dynamics at the global scale. The river network used in this study is a 3" flow direction map derived from the MERIT DEM (Yamazaki et al., 2017) and water body datasets including the Global 1" Water Body Map (G1WBM; Yamazaki et al., 2015), Global Surface Water Occurrence (GSWO; Pekel et al., 2016), and OpenStreetMap, which are referred to as MERIT Hydro (Yamazaki et al., 2019). The MERIT Hydro generation involved following steps. Initially a "conditioned DEM" was created by lowering the elevation of water pixels in MERIT DEM based on G1WBM, GSWO, and OpenStreetMap. Subsequently, an initial flow direction was determined based on topographic slope using "Steepest Slope Method". Some adjustments were made to ensure the flow continuity. Finally, endorheic basins were detected using Global 3" Water Body Map and Landsat tree density maps (Yamazaki et al., 2019). The MERIT Hydro include an adjusted DEM, river width, height over the nearest drainage, flow accumulation area, and flow direction data. The 3" MERIT Hydro was used to determine whether VSs were located on land, river, or ocean pixels. The allocation procedure for the higher-resolution flow direction map is described in Section 2.3."*

10) Lines 144-145: The authors should elaborate on how river bathymetry and river bank height cause deviations.

We thank referee #1 for the question. In CaMa-Flood hydrodynamic model the WSE is diagnosed using river bathymetry $(z - b)$ and riverbank elevations $(z)$.

$$WSE = d + z - b$$

Where $d$ is the river water depth, $z$ is the riverbank height, and $b$ is the river channel depth. Because of limited data availability for river bathymetry, a power-law relationship was employed to estimate the river channel depth. Additionally, riverbank elevations were derived from spaceborne DEMs that may have inherent errors. Consequently, any inaccuracies in the river bathymetry or riverbank elevation can significantly impact the calculations of WSE.

11) Section 3.1: The section lacks sufficient discussion.

We would like to express our gratitude for the suggestion. We have revised and updated the Section 3.1. Updated text were highlighted in *purple*.

*"3.1 Allocation of VSs to the river network*

*Figure 4a shows the global distribution of flags 10, 20, 30, and 40, which VSs initially located on river channel river, land with a single-channel river nearby, land with a multi-channel river nearby, and ocean pixels, respectively. Flag 10 was the most common, accounting for 71.74% of all VSs, followed by flags 20 (26.88%), 30 (1.34%), and 40 (0.04%). Flags 10 and 20 were evenly distributed worldwide. Mostly, large rivers such as Amazon, Congo, Nile, Ob, etc. consist of flags 10 or 20 which indicate the low inconsistencies between VS locations and the river network. Flag 40 is distributed near the ocean in Congo River, Santee River in United States, Lumi Semanit River in Albania, Mahavavy River in Madagascar, and Luni River in India. In addition, flag 30 can be seen mostly in mid-streams where multi-channel rivers exist. Hence, different flags shows different geographical characteristics.*

*The log probability distributions of upstream catchment areas for different flag values are also shown in Figure 4b. The median upstream catchment areas were $2.73 \times 10^4$, $9.95 \times 10^3$, $2.16 \times 10^4$, and $3.95 \times 10^4$ km² for flags 10, 20, 30, and 40, respectively. Flag 40 represented the largest median upstream catchment area because those are closer to the ocean and have large upstream catchment area. The distribution of flag 40 was strongly right skewed, influenced by the larger upstream catchment area of downstream Congo River. Flag 20 had the smallest median upstream catchment area, which indicates that most flag 20 VSs were in upstream reaches.*

*Figure 4c depicts the probability distribution of riverbank elevation for each flag. Lines represent the probability distributions of elevation for flags 10 to 30, with median values of 112.9m, 147.0m, and 141.2m for flag 10, flag 20, and flag 30, respectively. Notably, flag 40 was not visible in Figure 4c due to its very low elevation, with a median of 0.0m (mean=0.54m and std=1.21m). Flags 10 to 30 were distributed from mean sea level to 4790.0m, and there was no significant difference in elevation observed among flags 10 to 30.*

*The river width distribution for each flag is demonstrated in Figure 4d. Flag 20 exhibited the smallest median river width at 41.4m, with a relatively low standard deviation of 193.3m. On the other hand, Flag 40 displayed the largest median river width of 224.0m, but its variation was substantial (std=1336.6m) due to the wider Congo downstream, which measures around 3170.0m. Flag 10 showed a median river width value of 222.0m, comparable to Flag 40, but with a lower variation (std=683.6m). Meanwhile, Flag 30 exhibited a median river width of 77.7m, falling between the median river widths of Flag 10 and Flag 40. The large variation in river width observed for Flag 10 was due to its widespread distribution across the rivers, while the substantial variation of Flag 40 was influenced by the VSs' location in the Congo River."*

12) Figure 4: The authors should improve the overall quality of the figures, and in the density distribution plot, they should change the color code to make the density distribution line for Flag 10 visible.

Thanking the referee #1, we have improved the figures as shown below.

[Figure]

*Figure 4: Global map of allocation flags. Panel at lower left corner shows probability distribution of the upstream catchment area in log scale for different flags. Flags 10, 20, 30, and 40 are indicated by light blue, medium blue, dark blue, and red colors, respectively.*

13) Figure 5: The authors should improve the overall quality of the figures. The y-axis level is missing, and there is overlap between Figures 5a and 5b.

We would like to thank the referee #1. We have improved quality of Figure5, and a common y-axis label was added to Figures 5b for 5b, 5c, and 5d.

[Figure]

*Figure 5: a) Global distribution, b) histogram of catchment area (km2), c) histogram of elevation (m), and d) histogram of river width (m) of biased VSs. Light blue circles, medium blue diamonds, dark blue squares, and red triangles for flags 10, 20, 30, and 40, respectively in panel a.*

14) Line 228: The authors should explain why they compared the evaluated results in terms of RMSE? please consider showing the correlation coefficient and bias as well.

We express our gratitude for referred #1 for the valuable suggestions. We use RMSE because it represents overall nature of the errors and useful in evaluating the WSE against satellite altimetry. We have added correlation coefficient and bias to the Table 3

*"Table 3: Median statistics of the error of simulated WSE using CaMa-Flood hydrodynamic model. RMSE (root mean squared error), Bias, and CC (correlation coefficient) were presented. The simulated WSE is compared with HydroWeb satellite altimetry data where the VS s were allocated using AltiMaP or the ordinary allocation method."*

|  | AltiMaP | | | Ordinary | | |
|---|---|---|---|---|---|---|
|  | *RMSE* | *bias* | *CC* | *RMSE* | *bias* | *CC* |
| *All* | *2.68* | *-0.01* | *0.67* | *2.98* | *-0.99* | *0.67* |
| *Flag 10* | *2.65* | *-0.43* | *0.67* | *2.94* | *-1.87* | *0.68* |
| *Flag 20* | *2.71* | *-0.17* | *0.66* | *3.06* | *-2.46* | *0.66* |
| *Flag 30* | *2.72* | *-0.60* | *0.64* | *2.85* | *-1.97* | *0.61* |
| *Flag 40* | *0.85* | *-0.37* | *0.02* | *0.94* | *-0.30* | *0.02* |

15) Line 229: The authors should explain why the elevation causes an increase in RMSE.

Thank you for the question. In rivers at higher altitudes, the internal slope of the unit-catchment is more pronounced, resulting in greater height variation within the model grid compared to rivers at lower elevations. As a consequence, even VSs situated closer to the unit-catchment mouth can exhibit larger elevation biases compared to the river grid in lower elevation areas.

16) Line 230: The authors should rewrite the sentence to clarify that there is no change in RMSE before a certain threshold (<200 m/km) and that the medium of RMSE increases from 2 to 4 m, not just RMSE.

Thanking the referee #1, we revised the text as follows. The revisions are shown in purple:

*"As the distance from the VS location to the unit catchment mouth increased, the median RMSE of simulated WSE increased (Figure 6), mainly due to the difference in elevation between these points. Thus, large errors may be associated with simulated WSE when the VS is located far from the unit catchment mouth. Similarly, the median RMSE of simulated WSE increased slightly as the slope within the unit-catchment increased until slope < 200 m/km, with larger slopes (> 200 m/km) showing an increase in median RMSE from 2 to 4 m. This variation may have been caused by the non-uniformity of slopes within unit catchments of the CaMa-Flood model; however, it was well within the range of variation within unit-catchment slope bins, which reached up to 8 m."*

17) Line 265: The authors should explain why?

We would like to thank the referee 1# for asking for the clarification. Firstly, the RMSE was similar in global scale because flag 10 have more than 70% which may not contribute for the large error due to allocation method. Most Flag 20-40 would account for the differences in RMSE between AltiMaP and ordinary method. Secondly, there were VSs with lower RMSE in ordinary method than AltiMaP may be due to compensating for error due to other reasons such parameter errors in the model.

We have revised the text as follows:

*"RMSEs were calculated for WSEs simulated by CaMa-Flood and forced by VIC BC runoff (Lin et al., 2019). The spatial distributions of WSE RMSEs for VS allocations obtained using AltiMaP and the traditional method of allocating VSs to the CaMa-Flood grid are shown in Figure 7. Traditional VS allocation was performed using directly converting longitude and latitude information to coarse-resolution (i.e., 0.1°) grids. At the global scale, RMSEs were generally similar between both VS allocation methods. However, the satellite altimetry was better represented by AltiMaP for 17.52% of VSs (negative ΔRMSE) and by the traditional method for only 12.85% of VSs (positive ΔRMSE). The lower ΔRMSE of ordinary method may be due to the fact allocation to a nearby grid by ordinary method compensate for the errors in the model such as river bathymetry error (Modi et al., 2022)."*

18) Figure 7: The authors should discuss the expert method and ordinary method in the text to help readers better understand the differences between them. Alternatively, they could use constant terms to avoid confusion between the traditional method and ordinary method terms.

Here expert method simply refers to the method developed in this manuscript (AltiMaP). Thanking the referee, we revised all the instance of expert method to AltiMaP.

19) Figure 8: The authors should address the same comment as Figure 7. Additionally, they should move Figure 8 from outside of Table 3 to improve the organization of the paper.

We appreciate the referee #1 for pointing out unvolunteered error. We have revised the expert method to AltiMaP and move Figure 8 outside the Table 3.

---

## Author Comment (AC2)

**Anonymous Referee #2 - Comment on essd-2022-438**

We would like to express our gratitude to Referee #2 for meaningful comments and suggestion. We have carefully reviewed your comments and have made necessary updates to our manuscript. We provide point-to-point response to the referee comments shown in blue whereas the revision made to the main text in shown *blue Italics*.

Kind regards

This study presents a method called an automated altimetry mapping procedure (AltiMaP) that allocates altimetry virtual stations (VS) to the Multi-Error Removed Improved Terrain Hydrography (MERIT Hydro) river network. Although this study shows an improvement over the traditional method of allocating VS to the coarse-resolution river network, I have some questions and suggestions to the authors.

We are grateful to the referee #2 for valuables comments and suggestions.

1.      First of all, the authors are using the simulated WSEs from the CaMa-Flood model to be compared with the WSEs from altimetry with AltiMaP. I am not sure how this approach (i.e., using simulated WSEs as a reference to evaluate observed WSEs) can be convincing, especially considering the fact that the CaMa-Flood is a global hydrodynamic model calibrated/validated with available in-situ network (I assume so) which are sparse in large river basins, such as Amazon, Congo and Mekong. In addition, how AltiMaP assigned altimetry-observed WSEs can be used in the future for better calibration/validation of the CaMa-Flood model (this could be added in the summary section or in a separate discussion section)?
Can authors elaborate on this?

We extend our gratitude to referee #2 for providing insightful comments. We wish to clarify that our objective in comparing with CaMa-Flood results is to demonstrate how utilizing the AltiMaP method (developed in this manuscript) can enhance the comparison of simulated water surface elevation (WSE) with satellite altimetry. Our intention is not to evaluate the CaMa-Flood model itself. Rather, we employ CaMa-Flood as an illustrative example of model simulations to discuss how a systematic allocation of VSs can lead to improved model-satellite altimetry comparisons.

It is crucial to emphasize that we do not intend to use CaMa-Flood simulations as a reference; instead, we aim to evaluate the AltiMaP and ordinary allocation methods using the same dataset for both observations and simulations. In this study, we did not undertake an extensive calibration process for CaMa-Flood; instead, we utilized the standard parameters. In response to referee #2's first part of the comment, we have revised the manuscript, incorporating the additional text shown in purple.

*"We forced the CaMa-Flood hydrodynamic model using the runoff simulated by the Variable Infiltration Capacity (VIC) LSM (Liang et al., 1994) with bias correction (VIC BC) (Lin et al., 2019). The standard model parameters were used in this simulation including parameters such as river bathymetry, river width, and Manning's coefficient. For comparison with WSEs simulated by CaMa-Flood, we mapped VSs to a 6′-resolution global river network after allocating VSs to the*

*MERIT Hydro network at 3″-resolution using AltiMaP, because the CaMa-Flood river map was derived by upscaling the MERIT Hydro flow direction map using FLOW algorithm (Yamazaki et al., 2009). Then we compared the resulting simulated WSEs with observed WSEs mapped onto the river network based on the MERIT Hydro using the AltiMaP algorithm and the ordinary allocation method, i.e., converting longitude and latitude to the CaMa-Flood grid. In this evaluation, our primary objective is to assess the potential improvement brought about by the AltiMaP method when comparing simulated WSE with the ordinary allocation method. For a fair and unbiased evaluation, we employ the same dataset for both observations (i.e., satellite altimetry) and simulations. By doing so, we create a consistent and controlled environment to assess the performance of the AltiMaP method in comparison to the ordinary allocation method. We would like to emphasize that our intention is not to treat the CaMa-Flood simulation results as an absolute reference. Rather, we utilize them as a basis for evaluating the allocation methods concerning satellite altimetry data. Our aim is to investigate whether the AltiMaP method offers any notable advancements in the accuracy of simulated WSEs when compared to satellite-derived measurements."*

Secondly, as the referee #2 correctly mentioned one of the objective of the mapping of the VS into MERIT Hydro is to use them for calibration of model parameter (e.g., Zhou et al., 2022), correct state variables using assimilation (e.g., Revel et al., 2023), and model evaluation (Modi et al., 2022). The river bottom elevation parameter was corrected using a rating curve method by using both satellite altimetry and in-situ river discharge data (Zhou et al., 2022). The model can evaluate and calibrate using in-situ discharge, satellite altimetry, and inundation extent (Modi et al., 2022). In addition, the satellite altimetry data can assimilated into the model to correct the state variable (Revel et al., 2023). All the above were implemented to CaMa-Flood model and used the AltiMaP for them. We revised the text as below as per the referee #2's suggestion.

*"Because we used river network-related variables in the AltiMaP VS allocation algorithm, we were able to calculate distances and elevation differences between each VS and the unit-catchment river mouth. These parameters are particularly important for comparing WSEs simulated by coarse-resolution, large-scale river routing models such as CaMa-Flood, which are based on discretized river reaches with a representative elevation for each pixel. Minimizing the distance and elevation difference between the VS and unit-catchment river mouth is critical for improving the accuracy of WSE simulations. Thus, this elevation difference may be used as a proxy to interpret bias between simulated and observed WSEs (Fassoni-Andrade et al., 2021). Satellite altimetry data are also extremely useful for evaluating and calibrating hydrodynamic models (e..g., Zhou et al., 2022) and correcting variables through data assimilation (e.g., Revel et al., 2023b), which requires correct VS allocation to a river network map. The river bathymetry parameter can be calibrated using rating curve method developed using satellite altimetry and in-situ river discharge data (Zhou et al., 2022). Furthermore, the model can be evaluated using multi-variables (i.e., river discharge, WSE, and inundation extent) (Modi et al., 2022)."*

2.      Line 141: how is this assumption valid? It is not certain whether the observed WSE time series available from the HydroWeb is really the one over the floodplain (which could be flag 20 or 30), or over the open river channel. If the HydroWeb time series are indeed for the floodplain, AltiMaP may be erroneously assigning the VS to the open river channel. Can authors elaborate on this?

We would like to express our gratitude to the referee #2. Since the satellite altimetry can be successfully retrieved river wider than 0.8 km (Birkett and Beckley, 2010), we think the above assumption is valid in the case of the retrackers used in HydroWeb (ICE-1, ICE-2, etc) or other similar datasets even though some other methods can be useful for deriving WSE in narrow rivers (e.g., Sulistioadi et al., 2015).

Moreover, we provided the details about the secondary location (the small channel) as kx2 and ky2. Therefore, the users of the dataset can have flexibility to text both locations. This assumption is mostly used for the flag 30 and the amount of VS categorized as flag 30 is only 1.34% of the total VSs.

We analyzed the root mean squared error (RMSE) and bias using original and secondary location as shown in the Figure S2. We found that our assumption is mostly correct as the RMSE and biases of original location (kx1, ky1) is lower than those from the secondary location (kx2, ky2).

[Figure]

*Figure S2: Comparison of root mean squared error (RMSE: a) and bias (b)for Original and Secondary allocations.*

We believe the referee is referring to the floodplain lakes in "HydroWeb is indeed for floodplain" because radar echo will be contaminated from the ground (backscatter is useful from a smooth surface). Usually, HydroWeb defined the satellite altimetry over rivers and lakes separately. Therefore, if HydroWeb assign one VSs as river and provide a timeseries of floodplain lake means the observations were taken at non-nadir direction and those VS needed to correct or remove from the datasets. Checking the biased VS against MERIT DEM elevation can one way to identify such erroneous VSs. But to understand error is due non-nadir view or other issues need further investigation. In addition, MERIT Hydro incorporate waterbody map derived from Landsat and river flow direction delineated from MERIT DEM. Hence, most of the river permanent water areas were considered in the MEIRT Hydro.

Considering the referee #2's comment we added following discussion to the manuscript.

*"4.4 Limitations and Future Perspectives*
*Even though AltiMaP is suitable in mapping the VSs into the given river network with D8 connection, the method is not capable of identifying non-nadir observations (such as floodplain lakes near the river channel). One of the major problems in the conventional altimeters in low-resolution mode (LRM) such as ENVISAT was correcting the observations from the non-nadir view was treated as nadir observations (Calmant et al., 2008; Frappart et al., 2006; da Silva et al., 2012). The dual antenna configuration of the CryoSat-2 allows precise position of reflecting point in the radar footprint and solve the signal location along-track and across-track directions (Cretaux, 2022). Moreover, ICESat-1/2 data can also be a great source of importance over terrestrial waters, but the longer revisit time limit the applications in hydrology. Satellites such as CroySat-2 and ICESat-2 provide an addition challenge in using them in river monitoring. CryoSat-2 with its' drifting orbit ~7.5km makes it challenging to define VSs as in repeat orbits (Schneider et al., 2017). With the complex ground track configuration of ICESat-2 makes it complex to use in river monitoring because the assigning method would differ depend on the satellite track orientation with respect to the river centerline (Scherer et al., 2023). However, with slight modification to the AltiMaP, We would be able to map such data into the MERIT Hydro."*

3.      It is mentioned that mean observed WSEs are used to be compared with MERIT DEM elevation. But it is not explained how "mean" has been obtained. Did the authors simply take the mean of the entire WSE time series? Or did the authors consider the water cycle of the basins? If the entire time series has been simply used to compute the means, that will lead to an inherent bias due to the seasonality of WSE changes. Please clarify.

Thank you very much for the comment. We used simply used all the observation for simplicity for calculating the mean. As we compare the mean WSE with the maximum variation of WSE (30m), the seasonal bias can be smaller than the amplitude of the WSE pulse. Therefore, we believe the bias due the seasonality can be ignored. But we firmly believe it can be considered in the future.

Moreover, the users of the dataset can easily modify this condition as the filtering is separated from the allocation/mapping process.

4.      Figure 7: Even using AltiMaP, I see majority of the VSs have high RMSEs over the world. This demonstrates that basically HydroWeb WSEs and CaMa-Flood WSEs are not comparable. There are many factors behind this (as authors mentioned them), but I think the authors should not simply use the time series from HydroWeb without quality check. I'm not saying HydroWeb data is inaccurate, but I'm saying some of their time series may be inaccurate because of the inherent limitation of altimetry over land.

We firmly agree with the referee #2 that the quality checks must be implemented before using satellite altimetry before using them for any calibration/validation/assimilation purposes. Here, we introduce a simple quality check by comparing with the MERIT DEM elevation. However, more sophisticated quality control methods can be considered in the future studies.

In addition, we can get a good understanding which of the VS can be used with large-scale hydrodynamic model by the data provided in the AltiMaP dataset such as distance to the unit-catchment mouth (dist_to_mouth). By considering the only the VSs near to the unit-catchment mouth one make fare comparison with WSE simulated by CaMa-Flood.

 Minor comment:

1.      Abstract: "much lower (10.6%)" is a bit of exaggeration in my opinion. I would say "a meaningful improvement" or something like that.

Thanking the referee #2 we revised it.

Reference:
1.  Modi, P., Revel, M. and Yamazaki, D.: Multivariable Integrated Evaluation of Hydrodynamic Modeling: A Comparison of Performance Considering Different Baseline Topography Data, Water Resour. Res., 58(8), 1–20, doi:10.1029/2021WR031819, 2022.
2.  Revel, M., Zhou, X., Yamazaki, D. and Kanae, S.: Assimilation of transformed water surface elevation to improve river discharge estimation in a continental-scale river, Hydrol. Earth Syst. Sci., 27(3), 647–671, doi:10.5194/hess-27-647-2023, 2023.
3.  Zhou, X., Revel, M., Modi, P., Shiozawa, T. and Yamazaki, D.: Correction of River Bathymetry Parameters Using the Stage–Discharge Rating Curve, Water Resour. Res., 58(4), 1–26, doi:10.1029/2021WR031226, 2022.

---

## Author Comment (AC3)

**Anonymous Referee #3 - Comment on essd-2022-438**

We would like to express our gratitude to Referee #3 for meaningful comments and suggestion. We have carefully reviewed your comments and have made necessary updates to our manuscript. We provide point-to-point response to the referee comments shown in blue whereas the revision made to the main text in shown *blue Italics*.

Kind regards

General comments

This study develops an altimetry mapping approach (AltiMaP) that aims to mitigate the mismatches between virtual station (VS) locations and actual river locations, which are caused by DEM errors, the use of discrete river grids, and the allocation of VSs to the center of the WSE observation search area. The topic is interesting to the hydrological community. However, there are many issues unsolved with the manuscript in its present form and I recommend rejection (see details below).

We would like to extend our gratitude to the referee #3 for the time and effort to review our manuscript. We addressed all the comments of the referee #3 and the point-to-point answers were provided below.

1)      I think one of the major limitations of the study is that the allocation was performed based on Hydroweb whose VSs are located away from the actual river. What is the added value of the method for self-defined VSs or other data sets when the VSs are delineated right at the center of the river?

We extend our sincere gratitude to referee #3 for providing essential and valuable comments. We would like to reiterate that the primary objective of AltiMaP is to establish meaningful correspondences between VSs and MERIT Hydro, ultimately enabling a reliable comparison between model outputs and satellite altimetry data. While it is important to mention that the "correction of VS location" is indeed a step within the AltiMaP process, it is crucial to recognize that the true essence of AltiMaP lies in ensuring accurate VS-hydrography correspondences. AltiMaP not only facilitates the calibration of parameters but also plays a key role in effectively correcting states through data assimilation techniques.

We firmly believe that our method, AltiMaP, holds significant value even when VSs are self-defined, as demonstrated by the special case where VSs are located at the river mouth under the flag 10 (flag 13, as indicated in supplementary Table S1). In AltiMaP, we also delineate VSs at the centerline of the river, which corresponds to flag 10 in our convention (flag 11, see supplementary Table S1). Therefore, the cases highlighted by referee #3 are just special instances within our flagging approach in AltiMaP. We are confident that AltiMaP possesses broader applicability beyond these specific cases. It can be effectively utilized in various scenarios, making it a versatile and valuable tool for VS-hydrography correspondences and model-satellite altimetry comparisons.

*"Table S1: Secondary Flags used in the AltiMaP"*

| Main Flags | Secondry Flag | Description |
|---|---|---|
| | *11* | *VS was found on the river centerline* |
| *10* | *12* | *VS was found on the river channel but not in the centerline and assigned to the nearest centerline* |
| | *13* | *VS was found in the unit-catchment mouth* |
| *20* | *21* | *VS was found in the ground and assinged to the nearest single channel centerline* |
| | *22* | *VS was found in the ground near large river channel in in mult-channel river and assinged to the larger river centerline* |
| *30* | *31* | *VS was found in the ground near small river channel in mult-channel river and assinged to the large river centerline* |
| | *32* | *VS was found in bifuricating channel and assinged to the large river centerline* |
| *40* | *40* | *VS was found in the ocean and assinged to nearest river channel* |

Even though our objective in this study is to develop a robust methodology to map the VSs of the existing satellite altimetry products such as HydroWeb, DAHITTI, Hydrosat, Copernicus Global Land Service, the methods can be extended to self-defined VSs. We believe most of the WSE (satellite based or in-situ) observations with geographical information (lon/lat) can be mapped into the high-resolution river network map using the AltiMaP. For example, Schneider et al., (2017) projected the self-defined CryoSat-2 data into the model space. We were able to assign those data into MERIT Hydro using AltiMaP (Figure S1).

[Figure]

*Figure S1: AltiMaP allocation flags for the CryoSat-2 data provided by Schneider et al., (2017). Here each Cryostat-2 observations has been considered as a VS to allocate into MERIT Hydro because of the drifting orbit of CryoSat-2.*

Moreover, as deriving a global scale satellite altimetry dataset is challenging, for large-scale model such as CaMa-Flood comparison it may be inevitable to use existing datasets such as HydroWeb. According to our knowledge, none of the other models compared WSE against satellite altimetry on a global scale.

Considering the comment by the referee#3, we have revised the text following the referee #3 comment:

*"We introduce our automated altimetry mapping procedure (AltiMaP), which enable better evaluation of WSEs simulated by large-scale hydrodynamic models using available satellite altimetry data."*

On the other hand, despite of the VS self-defined or obtained from the organization, VS needed to be mapped to correct river grid or river reach in grid-based models. VS being point observation it needed to be compared against correct location. To understand the errors of the model simulations, it is important to know the relative position of VS within the unit-catchment. If the VS is delineated at unit-catchment mouth (similar to Flag 12 in Table S1), the observations can be directly compared with simulations.

In addition, we cannot expect the self-defined VS to be at the center of the river networks of the model which delineated using the spaceborne digital elevation model (DEM) which often presents with mismatches with the actual elevations due to vegetation bias, speckle noise, stripe noise, and absolute biases (Yamazaki et al., 2019). These errors in the DEMs can produce mismatches

between delineated river with ground truth. Moreover, the rivers can change the location with time (e.g., meandering). In such instances, the VS needed to be allocated to the correct river grid of the model river network.

2)      If the authors focus on satellite radar altimetry, the swath interferometric altimetry mission CryoSat-2 with dense spatial coverage should be added as an important data source for validating the method. Further, how applicable is your method to laser altimetry (ICESat-1/2)? I would provide a preliminary result for these data with a short discussion.

We would like to express our gratitude to referee #3 for his important comment. Of course, CryoSat-2 will provide unique data. CryoSat-2 is equipped with the Synthetic Aperture Interferometric Radar Altimeter (SIRAL), which operates in the Ku-band using synthetic aperture radar (SAR) mode. In addition, CryoSat-2 has the capability to perform SAR interferometric (SARIn) measurements using a dual antenna configuration. This SARIn mode allows for the accurate determination of the position of the reflecting point within the radar footprint. This feature is particularly valuable for identifying non-nadir measurements, which can occur when observing terrestrial waters. We think this identifying non-nadir measurements is beyond the scope of our study. We believe the data providers can use such method for quality control. We firmly believe that we should include CryoSat-2 based quality control in the future.

In addition, CryoSat-2 with its' drifting ground track a continuous river masks are needed (Jiang et al., 2017).  Moreover, we believe ICESat 1/2 is also provide important accurate estimates of WSE but with low resolution (~91 days). Both ICESat-2 and CryoSat-2 can be assigned to the river centerline depend on the orientation of the satellite track: 1) across-track and 2) along track (Scherer et al., 2022, 2023). Finding the nearest river centerline to the satellite footprint needs to the similar mapping procedure. We believe CryoSat-2 and ICESat-2 would be important addition to our dataset and useful in calibration and validation of global hydrodynamic models such as CaMa-Flood.

Considering the comments from the referee #3 we added some discussion to the manuscript as follows:

*"4.4 Limitations and Future Perspectives*
*Even though AltiMaP is suitable in mapping the VSs into the given river network with D8 connection, the method is not capable of identifying non-nadir observations (such as floodplain lakes near the river channel). One of the major problem in the conventional altimeters in low-resolution mode (LRM) such as ENVISAT was correcting the observations from the non-nadir view was treated as nadir observations (Calmant et al., 2008; Frappart et al., 2006; da Silva et al., 2012). The dual antenna configuration of the CryoSat-2 allows precise position of reflecting point in the radar footprint and solve the signal location along-track and across-track directions (Cretaux, 2022). Moreover, ICESat-1/2 data can also be a great source of importance over terrestrial waters, but the longer revisit time limit the applications in hydrology. Satellites such as CroySat-2 and ICESat-2 provide an addition challenge in using them in river monitoring. CryoSat-2 with its' drifting orbit ~7.5km makes it challenging to define VSs as in repeat orbits (Schneider et al., 2017). With the complex ground track configuration of ICESat-2 makes it complex to use in river monitoring because the assigning method would differ depend on the satellite track*

*orientation with respect to the river centerline (Scherer et al., 2023). However, with slight modification to the AltiMaP, We would be able to map such data into the MERIT Hydro."*

Moreover, we mapped CryoSat-2 water surface elevation from Schneider et al., (2017) over the Brahmaputra river (Figure S1). We treated each observation as a VS here because of the drafting ground track of the CryoSat-2. Then we projected the CryoSat-2 observations into the most appropriate nearest river pixels using AltiMaP which may slightly different from the method used by Schneider et al., (2017). Figure S2 shows the allocation flag map of CryoSat-2 data into the MERIT Hydro. We found 52.9%, 38.0%, and 9.1% of Flag 10, 20, and 30 but no Flag 40 was found because the data does not consist of the observations near the Ocean.

3)      Line 140: You are selecting the largest river for further processing. But it is possible that the observation (also termed POCA, point of closest approach) is from the river closest to the satellite (within the beam limited footprint) when there are multiple river channels near the VS location. Therefore, it would be interesting to perform a similar analysis with the abovementioned assumption (i.e., choosing the closest river as opposed to the largest river to derive WSE) to see the difference.

Thank you very much for the nice suggestion. The assumption that the observation is from the largest river when there are multiple river channels near the VS location due to the fact that the satellite altimetry can be derived only from rivers with substantial river width (e.g., 0.8 km) (Birkett and Beckley, 2010). Hence, we believe that the assumption is valid for the retrackers used in HydroWeb or other similar datasets even though some other methods can be useful for deriving WSE in narrow rivers (e.g., Sulistioadi et al., 2015).

In addition, we have already provided location of nearest smaller river location as kx2, ky2 in the detail dataset and geographic distance for VS location to two selected river pixels in the MERIT Hydro as dist1 and dist2 in AltiMaP dataset. We included a simple analysis using secondary locations kx2, ky2 (only for Flag 30).

[Figure]

*Figure S2: Comparison of root mean squared error (RMSE: a) and bias (b)for Original and Secondary allocations.*

4)	Line 148: Many previous studies have confirmed the reliability of the median value compared with the mean, which is quite sensitive to outliers. I would suggest the authors use the median value as the final WSE and update all the relevant results.

We agree with the referee that mean value can be affected by outliers. But as we are using pre-processed data such as HydroWeb where the outliers have been already removed, the mean values may not be corrupted. We check both mean and median and found that both the values are almost similar and works well for our purpose.

5)	Do you use lat and lon at nadir for the allocation of the altimetry measurements? If so, I guess you may need to use the corrected ones instead (lat_cor and lon_cor), which are better representations of the radar echoes.

Thank you for the question. We used the lat and lon information provided by the HydroWeb dataset. The lat/lon values represent the center location of the area allocated for the VS. We think the lat/lon locations provided in HydroWeb is different from locations of radar echoes.

6)	While range correction derived from waveform retracking is not within the scope of the manuscript, it is still one of the major sources of error for WSE. The introduction section should at least mention this. It would make more sense to briefly introduce the processing chain of Hydroweb (e.g., what retracker and/or slope correction method it is using), followed by a citation, such that authors without expertise in altimetry could better capture the contribution of the methodology.

Thanking the referee #3, we include additional details to the Table 1.

Minor comments

1)    Line 40: 'following troposphere'-> following dry troposphere?

Thanking referee #3, we revised it.

2)    Line 163: what is the timestamp of the MERIT DEM? Because your altimetry data cover a wide range of time periods (1992–2022), how could you confirm the MERIT DEM is representative of the actual topography that is validated against the altimetry missions?

Thank you very much for the question. MERIT DEM is derived from SRTM which was from 2000s and AW3D DEM which operated from 2006-2011 (Yamazaki et al., 2017). The DEM may not be the representative for each satellite mission. This highlights the importance of AltiMaP framework which can provide a mapping table for VSs into the MERIT Hydro river network which then feasibly used for simulation derived from MERIT Hydro.

3)    Line 184: 'then adding 100 to the flag of any VS that is biased', please explain or reword. Why not add a fifth flag for biased VSs?

We would like to keep the initial allocation flag preserved. Therefore, we added 100 to the existing flag to denote that as biased VS.

4)    Table 2: how to obtain the river widths, manually?

The river widths data was provided in MERIT Hydro dataset (Yamazaki et al., 2019). River widths were calculated using an algorithm developed by Yamazaki et al., (2014) using optical imagery.

5)    Line 202: 'river channel river'. A typo here?

Thank you for pointing this out. We carefully checked the manuscript for correct such typos.

6)    Figure 4: please increase the font size of the figure

Thank you very for the valuable suggestion. We have revised the Figure 4 as shown below.

[Figure]

*Figure 4: Global map of allocation flags. Panel at lower left corner shows probability distribution of the upstream catchment area in log scale for different flags. Flags 10, 20, 30, and 40 are indicated by light blue, medium blue, dark blue, and red colors, respectively.*

7)     Figure 5: please add a title for the y-axis in b,c, and d

Thanking the referee #3, we revised the Figure 5 as follows.

[Figure]

*Figure 5: a) Global distribution, b) histogram of catchment area (km2), c) histogram of elevation (m), and d) histogram of river width (m) of biased VSs. Light blue circles, medium blue diamonds, dark blue squares, and red triangles for flags 10, 20, 30, and 40, respectively in panel a.*

Reference:
1. Birkett, C. M. and Beckley, B.: Investigating the Performance of the Jason-2/OSTM Radar Altimeter over Lakes and Reservoirs, Mar. Geod., 33, 204–238, doi:10.1080/01490419.2010.488983, 2010.
2. Jiang, L., Schneider, R., Andersen, O. B. and Bauer-Gottwein, P.: CryoSat-2 altimetry applications over rivers and lakes, Water (Switzerland), 9(3), 1–20, doi:10.3390/w9030211, 2017.
3. Scherer, D., Schwatke, C., Dettmering, D. and Seitz, F.: ICESat-2 Based River Surface Slope and Its Impact on Water Level Time Series From Satellite Altimetry, Water Resour. Res., 58(11), doi:10.1029/2022WR032842, 2022.
4. Scherer, D., Schwatke, C., Dettmering, D. and Seitz, F.: ICESat-2 river surface slope (IRIS): A global reach-scale water surface slope dataset, Sci. Data, 10(1), 1–13, doi:10.1038/s41597-023-02215-x, 2023.
5. Schneider, R., Nygaard Godiksen, P., Villadsen, H., Madsen, H. and Bauer-Gottwein, P.: Application of CryoSat-2 altimetry data for river analysis and modelling, Hydrol. Earth Syst. Sci., 21(2), 751–764, doi:10.5194/hess-21-751-2017, 2017.
6. Sulistioadi, Y. B., Tseng, K. H., Shum, C. K., Hidayat, H., Sumaryono, M., Suhardiman, A., Setiawan, F. and Sunarso, S.: Satellite radar altimetry for monitoring small rivers and lakes in

Indonesia, Hydrol. Earth Syst. Sci., 19(1), 341–359, doi:10.5194/hess-19-341-2015, 2015.

7.  Yamazaki, D., O'Loughlin, F., Trigg, M. A., Miller, Z. F., Pavelsky, T. M. and Bates, P. D.: Development of the Global Width Database for Large Rivers, Water Resour. Res., 50(4), 3467–3480, doi:10.1002/2013WR014664, 2014.

8.  Yamazaki, D., Ikeshima, D., Tawatari, R., Yamaguchi, T., O'Loughlin, F., Neal, J. C., Sampson, C. C., Kanae, S. and Bates, P. D.: A high-accuracy map of global terrain elevations, Geophys. Res. Lett., 44(11), 5844–5853, doi:10.1002/2017GL072874, 2017.

9.  Yamazaki, D., Ikeshima, D., Sosa, J., Bates, P. D., Allen, G. H. and Pavelsky, T. M.: MERIT Hydro: A High-Resolution Global Hydrography Map Based on Latest Topography Dataset, Water Resour. Res., 55(6), 5053–5073, doi:10.1029/2019WR024873, 2019.

10.

---

## Author Response (AR2)

Dear Handling Editor and Referees,

We want to convey our appreciation for the valuable feedback and constructive comments received. Your insightful input and suggestions have enriched the quality of our manuscript. We are grateful for the time and effort you invested in reviewing our work. We provided point-to-point response to the referee comments shown in blue whereas the revision made to the main text in shown *blue Italics*. The revisions are highlighted in purple.

Kind regards

**Referee #4 - João Paulo Brêda**

I want to congratulate the authors for this work. This study describes the development of a dataset given by the allocation of virtual stations to the MERIT Hydro river centerline. In addition, the authors cross-validated the WSE simulated by a global hydraulic model (CaMa-Flood) built from the MERIT-Hydro dataset and the VS observations. To be honest, the authors did not present any great novelty with this approach for VS allocation, however, working with such big data on a global scale and the posterior analysis deserves a publication. Also, the dataset is going to be useful, especially for the next applications of global models built from the same MERIT-Hydro dataset.

We would like to thank the Referee #4 for the valuable comments and suggestions.

I just have a few comments that the authors could consider before publishing:

1. Are both datasets (MERIT and VS) referenced to the same geoid?

   We would like to thank the Referee #4 for question. Water surface elevations (WSEs) from satellite altimetry were converted to EGM96 before comparing MERIT and VS.

   Considering the Referee #4's comments, we revised text is shown in purple as follows:

   *"RMSEs were calculated for WSEs simulated by CaMa-Flood and forced by VIC BC runoff (Lin et al., 2019). Both simulations and observations were converted to the same geoid before calculating RMSE (i.e., EGM96). The spatial distributions of WSE RMSEs for VS allocations obtained using AltiMaP and the traditional method of allocating VSs to the CaMa-Flood grid are shown in Figure 7. Traditional VS allocation was performed using directly converting longitude and latitude information to coarse-resolution (i.e., 0.1°) grids. At the global scale, RMSEs were generally similar between both VS allocation methods. However, the satellite altimetry was better represented by AltiMaP for 17.52% of VSs (negative ΔRMSE) and by the traditional method for only 12.85% of VSs (positive ΔRMSE) The lower ΔRMSE of ordinary method may be due to the fact allocation to a nearby grid by ordinary method compensate for the errors in the model such as river bathymetry error (Modi et al., 2022)."*

2. How does the algorithm automatically classify nearest multi-channel and nearest single-channel? It is not specified in the manuscript. (I assume that the rivers -flagged 10- are indicated on the MERIT-Hydro itself).

We would like to thank the Referee #4 for the valuable comment. The AltiMaP algorithm identifies river sections perpendicular to a given river, assessing their downstream connectivity. In addition, we defined a distance threshold in perpendicular direction to identify only multi-channel rivers.

Considering the referee's comments we modified the flowing (updated text in purple)

*"VSs must be assigned to river network pixels of the hydrodynamic model for accurate comparison of simulated and observed WSEs. The DEM-based river network can deviate from the cause of the actual river due to errors in DEM and low representability of the coarse-resolution of the river network (Amatulli et al., 2022; Paz et al., 2006; Yamazaki et al., 2009). Moreover, the reported location of the VS provided in HydroWeb can be further away from the actual river because HydroWeb provides the center of the search region, within a range of a few kilometers (e.g., 5 km × 5 km). Therefore, an important step in allocating VSs to large-scale hydrodynamic models is to assign each VS to a river centerline on a higher-resolution flow direction map (e.g., MERIT Hydro, at 3"). A schematic diagram of this allocation process is shown in Figure 1. Initially, the satellite altimetry auxiliary data (e.g., longitude and latitude) for each VS were converted into 3" pixels. Then we flagged each VS according to the land cover of the initial allocation of the pixel, with 10, 20, 30, and 40 representing river channel, land with the nearest single-channel river, land with the nearest multi-channel river, and ocean pixels, respectively (Figure 1). The secondary flags also defined to represents more special cases as defined supplementary Table S1. Finally, we searched for the centerline of the nearest river according to geometric distance and allocated the VS to that location. VSs initially located on land pixels with the nearest multi-channel rivers were allocated to the nearest largest channel of the multi-channel river (considering the upstream catchment area). The* *AltiMaP identifies multi-channel river by searching in a direction perpendicular to the specified river considering their downstream connectivity. We assume the observation is from the largest river when there are multiple river (Supplementary Figure S1) channels near the VS location because backscatter from the narrow river can be highly influenced by non-water features and mostly successful retrievals of WSE can be seen on larger rivers than ~0.8 km. (Birkett et al., 2002)."*

In the AltiMaP algorithm, if the initial allocation of VSs coincides with a MERIT Hydro-defined river, the algorithm will define it as flag 10.

3. Line 194. Just an observation: The criteria for the removal of a VS should be also related to the standard deviation of the respective VS (or the maximum water level difference). If the water level doesn't vary more than a few meters (1 or 2) annually it doesn't make sense to keep a VS that its mean is more than 10 m higher or lower than the actual DEM.

We would be grateful for the Referee #4 for the valuable suggestions. We also think that the standard deviation should also be considered in some instances depending on the application

of the satellite altimetry data. But in this study, we try to keep the comparison simple as possible by using MERIT elevation only as reference on the standard deviation for WSEs are hard to find.

Secondly, the removal of biased VS is a post-processing of the AltiMaP. Hence, we keep room for the users to tailor their requirements in removing biased VSs. Depending on application the users have the flexibility to change the criteria for filtering biased VS.

In addition, this kind of low variation (e.g., 1-2m) of WSE can be mostly due to non-nadir direction observations. Correcting such kind of errors are beyond the scope of our study but we have discussed this in the section "4.4 Limitations and Future Perspectives".

4. Line 309. Verb missing: "may be very small"?

Thanking the referee, we corrected it.

5. Line 413. Is this "min_val, max_val" supposed to be there?

Thank you very much pointing out this mistake. "min_val, max_val" should be removed from this text.

**Anonymous Referee #2**

We express our gratitude for Referee #2 for the time and effort dedicated to our manuscript.